

# Turbulence in a coastal environment: the case of Vindeby

Rieska Mawarni Putri[a], Etienne Cheynet[b], Charlotte Obhrai[a], and Jasna Bogunovic Jakobsen[a]

[a]Department of Mechanical and Structural Engineering and Materials Science, University of Stavanger, N-4036 Stavanger, Norway
[b]Bergen Offshore Wind Centre (BOW) and Geophysical Institute, University of Bergen, 5007 Bergen, Norway

**Correspondence:** Rieska Mawarni Putri (rieska.m.putri@uis.no)

**Abstract.** Turbulence spectral characteristics for various atmospheric stratifications are studied using the observations from an offshore mast at Vindeby wind farm. Measurement data at $6\,\mathrm{m}$, $18\,\mathrm{m}$ and $45\,\mathrm{m}$ above the mean sea level are considered. At the lowest height, the normalized power spectral densities of the velocity components show deviations from Monin-Obukhov similarity theory (MOST). A significant co-coherence at the wave spectral peak frequency between the vertical velocity

component and the velocity of the sea surface is observed, but only when the significant wave heights exceed $0.9\,\mathrm{m}$. The turbulence spectra at $18\,\mathrm{m}$ generally follow MOST and are consistent with the empirical spectra established on the FINO1 offshore platform from an earlier study. The data at $45\,\mathrm{m}$ is associated with a high-frequency measurement noise which limits its analysis to strong wind conditions only. The estimated co-coherence of the along-wind component under near-neutral atmosphere matches remarkably well with those at FINO1. The turbulence characteristics estimated from the present dataset are

valuable to better understand the structure of turbulence in the marine atmospheric boundary layer and are relevant for load estimations of offshore wind turbines. Yet, a direct application of the results to other offshore or coastal sites should be exercised with caution, since the dataset is collected in shallow waters and at heights lower than the hub height of the current and the future state-of-the-art offshore wind turbines.

# 1 Introduction

In the early 1990s, the first generations of offshore wind farms were commissioned to test the viability of extracting wind power in the marine atmospheric boundary layer (MABL). The first was the Vindeby Wind farm which provided electricity to around 2,200 homes during its 25 years of operation, with a total generated power of 243 GWh (Power Technology, 2020). The project was deemed successful and marked the beginning of the offshore wind sector.

Not only was the Vindeby project the first offshore wind farm, but it also provided precious information on meteorological conditions in the MABL using offshore and onshore meteorological masts. The data collected has been used to study the characteristics of the mean wind speed profile under various atmospheric conditions (Barthelmie et al., 1994; Barthelmie, 1999).



The masts were also instrumented with 3D sonic anemometers to study turbulence, but these data were used in a limited number of studies only (e.g. Mahrt et al., 1996, 2001).

The characteristics of the MABL differ from the overland atmospheric boundary layer (ABL) due to the larger proportion of the occurrence of non-neutral atmospheric stability conditions than on land (Barthelmie, 1999; Archer et al., 2016). Since the 2010s, several studies have indicated that diabatic wind conditions may significantly affect the fatigue life of offshore wind turbines (OWTs) components (Sathe et al., 2013; Hansen et al., 2014; Holtslag et al., 2016; Doubrawa et al., 2019; Nybø et al., 2020; Putri et al., 2020). Recent measurements from the first commercial floating wind farm (Hywind Scotland) have even

shown the direct influence of the atmospheric stability on the floater motion (Jacobsen and Godvik, 2021). Diabatic conditions are more likely to affect floating wind turbines than bottom-fixed ones as the first few eigenfrequencies of large floating wind turbines are close to or below $0.20\,\mathrm{Hz}$ (Nielsen et al., 2006), which is the frequency range mainly affected by the thermal stratification of the atmosphere. To model properly the wind load for wind turbine designs, a better understanding of the spectral structure of turbulence in the MABL is necessary, which addresses partly the first of the three great challenges in the field of

wind energy (Veers et al., 2019).

The limitations of current guidelines for offshore turbulence modelling, such as IEC 61400-1 (2005), have been highlighted in the past (Cheynet et al., 2017, 2018). Site-specific measurements advised by IEC 61400-1 (2005) are justified for the mean flow and integral turbulence characteristics. However, for the spectral characteristics, appropriate scaling can be used to display universal shapes over specific frequency ranges. In this regard, the present study addresses similar challenges as discussed by

Kelly (2018) but focuses on some specific aspects not covered by the spectral tensor of homogeneous turbulence, upon which the model by Kelly (2018) was developed: (1) the low-frequency fluctuations are generally underestimated by the uniform-shear model, especially under convective conditions (De Maré and Mann, 2014; Chougule et al., 2018) and (2) the vertical coherence of turbulence is not always described accurately by the spectral tensor (Mann, 1994; Cheynet, 2019).

Using the unexplored sonic anemometer data from the Vindeby database, this study looks at the characteristics of offshore

turbulence in the frequency space. The objective is to quantify the similarities between these characteristics and those identified on the FINO1 platform (Cheynet et al., 2018). Such a comparison is relevant to establish new offshore wind turbulence models that can be used to improve the design of the future multi-megawatt offshore wind turbines. Whereas the measurement data from the FINO1 platform were obtained $40\,\mathrm{km}$ away from the shore, at heights between $40\,\mathrm{m}$ and $80\,\mathrm{m}$ above the mean sea level (amsl), those from the Vindeby database were collected only $3\,\mathrm{km}$ from the seaside and altitudes between $6\,\mathrm{m}$ and $45\,\mathrm{m}$

amsl. Therefore, the two datasets offer a complementary description of wind turbulence above the sea.

The present study is organized as follow: Section 2 describes the instrumentation and the site topography. Section 3 summarises the data processing, the assumptions, and the models used to study the spectral characteristics of turbulence. Section 4 presents the methodology used to assess the data quality and selection of stationary velocity data. Section 5 first evaluates the applicability of surface-layer scaling for the anemometer records at $6\,\mathrm{m}$ amsl. Then, the one-point velocity spectra and co-coherence estimates

from Vindeby are compared with the semi-empirical models from the FINO1 platform to assess the similarities between both sites. Finally, the applicability of the Vindeby database for the design of an adequate turbulence model for offshore wind turbines is discussed in Section 6.



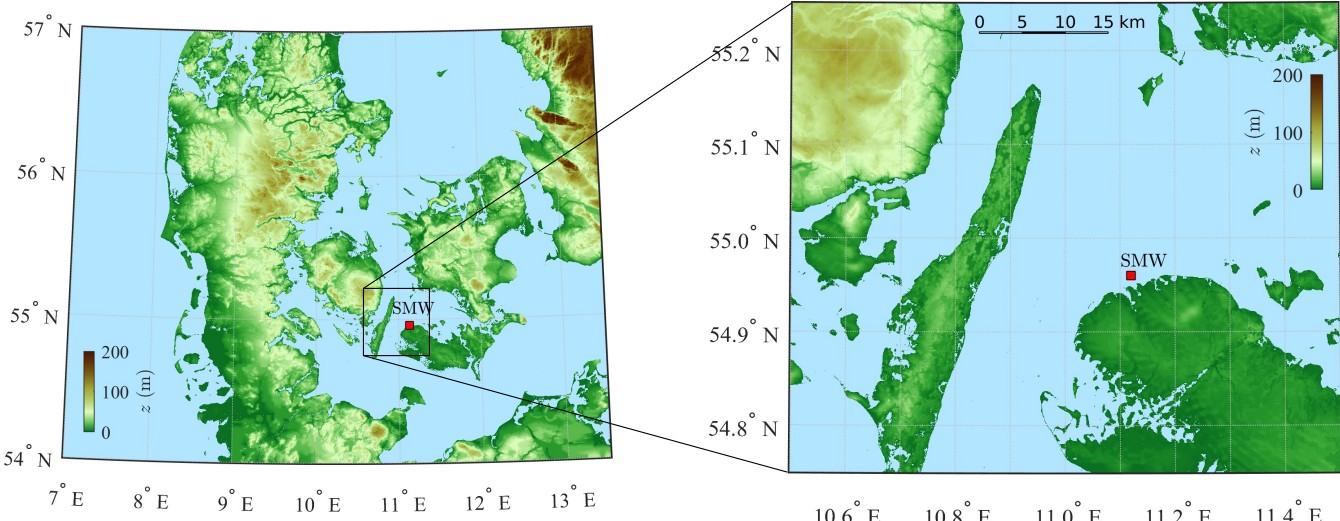

**Figure 1.** Digital elevation model of Southern Denmark showing the location of South Mast West (SMW), in a sheltered flat coastal environment.

## 2 Instrumentation and site description

Vindeby Wind Farm operated in Denmark from 1991 to 2016 and was decommissioned in 2017. It was located about $1.5\,\mathrm{km}$
to $3\,\mathrm{km}$ from the northwestern coast of Lolland Island (fig. 1). Due to its location, Vindeby may be regarded as a coastal site
instead of an offshore one. Vindeby has a flat topography with an average elevation under $11\,\mathrm{m}$ amsl, whereas the water depth
around the wind farm ranges from $2\,\mathrm{m}$ to $5\,\mathrm{m}$ (Barthelmie et al., 1994). As pointed out by Johnson et al. (1998), the average
significant wave height $H_s$ at Vindeby is under $1\,\mathrm{m}$. The water depth increases from ca.$3\,\mathrm{m}$ in the proximity of the wind farm up
to ca.$20\,\mathrm{m}$, away from the northern side of the wind farm.

The wind farm comprised of 11 Bonus $450\,\mathrm{kW}$ turbines arranged in two rows with $300\,\mathrm{m}$ spacing along the 325°-145° line
and three meteorological masts (fig. 2). The three masts were the Land Mast (LM), the Sea Mast South (SMS), and the Sea Mast
West (SMW) where the two latter were placed offshore (fig. 2). Both SMS and SMW were installed in 1993 and decommissioned
in 2001 and 1998, respectively. The present study considers only wind measurements from SMW due to the availability of
the data. Information on the measurement from LM and SMS linked to the atmospheric stability conditions can be found in
Barthelmie (1999).

The SMW was a triangular lattice tower with a height of $48\,\mathrm{m}$ amsl as sketched in fig. 3. The booms on the SMW were
mounted on both sides of the tower at 46° and 226° from the north and are referred to as the northern and southern boom,
respectively. The booms' length ranged from $1.6\,\mathrm{m}$ to $4.0\,\mathrm{m}$ and their diameter was $50\,\mathrm{mm}$ (Barthelmie et al., 1994). Three
F2360a GILL 3-axis ultrasonic anemometers (SAs) were mounted on the southern booms at $45\,\mathrm{m}$, $18\,\mathrm{m}$ and $6\,\mathrm{m}$ amsl and
operated with a sampling rate of $20\,\mathrm{Hz}$. Two Risø P2021 resolver wind vanes with wind direction transmitters P2058 were
located on the northern booms at $43\,\mathrm{m}$ and $20\,\mathrm{m}$ amsl using a sampling frequency of $5\,\mathrm{Hz}$. The height of the vanes and the cup

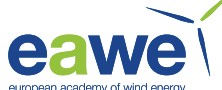


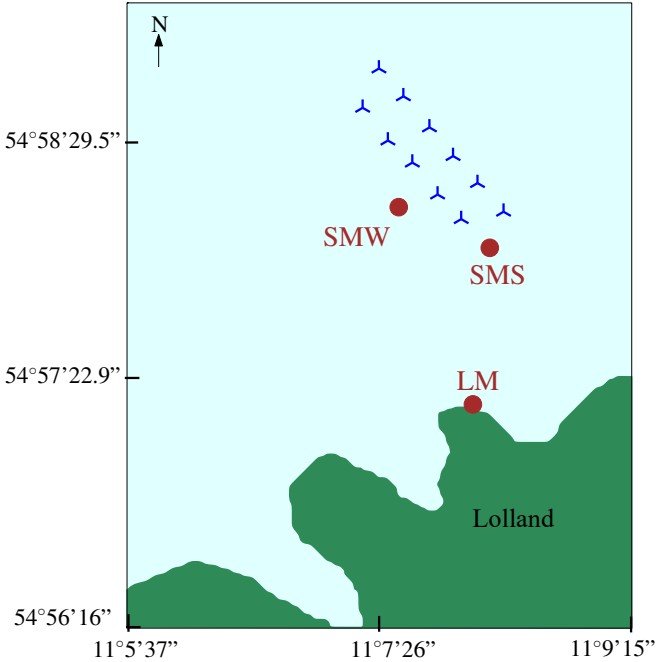

**Figure 2.** Vindeby Wind Farm layout with circles marking the position of the masts: SMW, SMS, and LM.

anemometers' centres above the boom was $600\,\text{mm}$. The air temperature at $10\,\text{m}$ amsl was recorded using a Risø P2039 PT 100 sensor. The sea surface elevation $\eta$ was measured using an acoustic wave recorder (AWR) placed on the seabed, $30\,\text{m}$ away from SMW, at a depth of $4\,\text{m}$ (Johnson et al., 1998). The sea surface elevation data was recorded at a sampling frequency of $8\,\text{Hz}$ but stored with a sampling frequency of $20\,\text{Hz}$.


The data collected from SMW were transferred to LM using an underwater fibre optic link and stored as time series of $30\,\text{min}$ duration. Such a duration is appropriate to study the wind turbulence at coastal and offshore areas (Dobson, 1981). Therefore, the flow characteristics studied herein are based on the averaging time of $30\,\text{min}$.

The fetch around SMW comprises of open sea, land, and mixed fetch as shown in fig. 1. The so-called sea fetch is considered

when the wind blows from $220°$ to $90°$, with a fetch distance up to $135\,\text{km}$ for the sector ranging from $345°$ to $355°$. The direction sectors from $0°$-$50°$ are those most affected by flow distortion due to the presence of the mast (Barthelmie et al., 1994). Furthermore, the flow from $335°$-$110°$ might be affected by the wake effects from the wind farm. To exclude flow disturbed by the presence of the mast and the wind turbines, only the flow from $220°$-$330°$ is considered in the present study, which represents 40% of the velocity data recorded in 1994 and 1995 at SMW. The surface roughness $z_o$ within $247°$ to $292°$ direction varies

with the mean wind speed from $1.1 \times 10^{-4}\,\text{m}$ to $1.2 \times 10^{-3}\,\text{m}$ (Johnson et al., 1998). A more detailed description of the other directional sectors is given by Barthelmie et al. (1994).





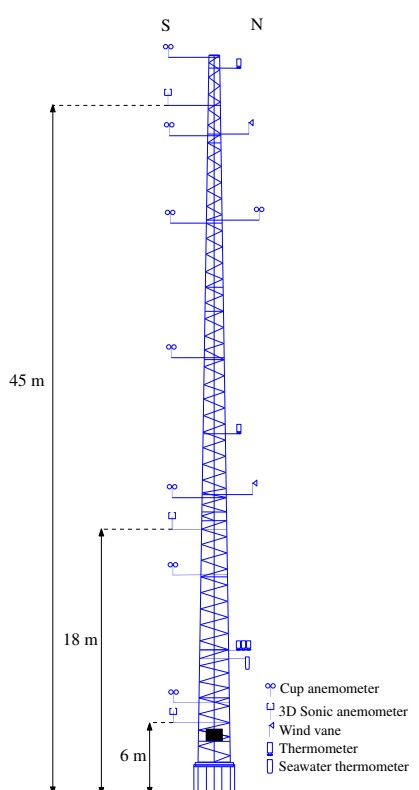

**Figure 3.** Instrument arrangement at SMW.

## 3 Theoretical background

### 3.1 Monin-Obukhov theory

The along-wind, cross-wind, and vertical velocity components are denoted $u$, $v$, and $w$, respectively. Each component is split
into a mean $(\overline{u}, \overline{v}, \overline{w})$ and fluctuating part $(u', v', w')$. In flat and homogeneous terrains, the flow is fairly horizontal, i.e. $\overline{v}$ and $\overline{w}$
are approximatively zero. To study turbulence for wind turbine design, the fluctuating components are assumed to be stationary,
Gaussian, ergodic random processes (Monin, 1958).

Although the $u$-component drives the wind turbine's rotor fatigue loads, proper modelling of the $v$-component may be
necessary for skewed flow conditions, which can occur because of a large wind direction shear (Sanchez Gomez and Lundquist,
2020) or wind turbine yaw error (Robertson et al., 2019). To estimate a wind turbine's fatigue loads, the vertical velocity
component is likely more relevant in complex terrain than offshore (Mouzakis et al., 1999). Nevertheless, this component is
studied here for the sake of completeness. Also, the vertical velocity component provides precious information on the sonic
anemometer flow distortion (Cheynet et al., 2019; Peña et al., 2019). The vertical velocity component is also necessary to assess





the atmospheric stability using the eddy covariance method and facilitates the study of the waves' influences on the velocity data
recorded by the sonic anemometers (e.g. Benilov et al., 1974).

In the surface layer, where MOST generally applies, the scaling velocity is the friction velocity $u_*$, whereas the scaling
lengths are the height $z$ above the surface and the Obukhov length $L$ (Monin and Obukhov, 1954), defined as

$$L = -\frac{u_*^3 \overline{\theta}_v}{g\kappa(\overline{w'\theta'_v})} \tag{1}$$

where $\overline{\theta}_v$ is the mean virtual potential temperature, $g = 9.81\,\mathrm{ms}^{-2}$ is the gravitational acceleration, $\kappa \approx 0.4$ is the von Kármán
constant (Högström, 1985), and $\overline{w'\theta'_v}$ is the flux of virtual potential temperature. For a given height $z$ above the surface, the
non-dimensional Obukhov length $\zeta = z/L$ is used herein to classify the thermal stratification of the atmosphere.

While $\theta'_v$ can be fairly well approximated by the fluctuating sonic temperature measurement (Schotanus et al., 1983;
Sempreviva and Gryning, 1996), the mean value $\overline{\theta}_v$ could not be reliably obtained from the sonic anemometers deployed on
SMW. Therefore, $\overline{\theta}_v$ was obtained using the absolute temperature recorded from the Risø P2039 PT 100 sensor at $10\,\mathrm{m}$ amsl,
which was converted into the virtual potential temperature using the pressure data from LM and assuming an air relative humidity
of 90% near to the sea surface (Stull, 1988). The air pressure data from LM is used due to the absence of air pressure data at
SMW and SMS.

Because the covariance between the cross-wind and the vertical component may not be negligible in the MABL (Geernaert,
1988; Geernaert et al., 1993), the friction velocity $u_*$ is computed as suggested by Weber (1999), that is

$$u_* = \sqrt[4]{\overline{u'w'}^2 + \overline{v'w'}^2} \tag{2}$$

A common approach to assess the applicability of MOST is to study the ratio $\phi_w = \sigma_w/u_*$ and the non-dimensional mean
wind speed profile $\phi_m$ defined as

$$\phi_m\left(\frac{z}{L}\right) = \frac{kz}{u_*}\frac{\partial\overline{u}}{\partial z} \tag{3}$$

as a function of the atmospheric stability (Kaimal and Finnigan, 1994). In the following, $\phi_w$ and $\phi_m$ are empirically modelled
such that following Kaimal and Finnigan (1994),

$$\phi_m \approx \begin{cases} (1 + 15.2|\zeta|)^{-1/4}, & -2 \leq \zeta < 0 \\ 1 + 4.8(\zeta), & 0 \leq \zeta \leq 1 \end{cases} \tag{4}$$

$$\phi_w \approx \begin{cases} 1.25\,(1 + 3|\zeta|)^{1/3}, & -2 \leq \zeta < 0 \\ 1.25\,(1 + 0.2|\zeta|), & 0 \leq \zeta \leq 1 \end{cases} \tag{5}$$

The validity of eq. (4) and eq. (5) is assessed for each anemometer in section 5.1. It should be noted that the presence of
waves, especially swell, may invalidate MOST in the first few meters above the surface (Edson and Fairall, 1998; Sjöblom and
Smedman, 2003b; Jiang, 2020) and this possibility will be discussed in section 5.4. Under convective conditions, the validity of





MOST may also be questionable if the fetch is only a few kilometres long due to the presence of internal boundary layers (Jiang et al., 2020). In the present case, the choice of wind directions from 220° to 330° limits strongly the possibility that internal boundary layers are affecting the velocity measurements.

### 3.2 One-point turbulence spectrum

The one-point velocity spectrum is a key quantity to model the dynamic wind load and the power production of wind turbines (Sheinman and Rosen, 1992; Hansen and Butterfield, 1993). One-point integral turbulence characteristics, especially the turbulence intensity, are not always appropriate for turbulence characterisation (Wendell et al., 1991) which motivates the study of the spectral characteristics of turbulence in the present study.

Following Kaimal et al. (1972), the normalized surface-layer one-point velocity spectra express a universal behaviour in the
inertial subrange

$$\frac{f S_u(f)}{u_*^2 \phi_\epsilon^{2/3}} \simeq 0.3 f_r^{-2/3} \text{ at } f_r \gg 1 \tag{6}$$

$$\frac{f S_v(f)}{u_*^2 \phi_\epsilon^{2/3}} \approx \frac{f S_w(f)}{u_*^2 \phi_\epsilon^{2/3}} \simeq 0.4 f_r^{-2/3} \text{ at } f_r \gg 1 \tag{7}$$

where $f_r = f z / \overline{u}$ and $f$ is the frequency; $S_u$, $S_v$, and $S_w$ are the velocity spectra for the along-wind, cross-wind, and vertical velocity component, respectively; $\phi_\epsilon$ is the non-dimensional turbulent kinetic energy dissipation rate (Wyngaard and Coté,
145 1971):

$$\phi_\epsilon^{2/3} = \frac{\kappa z \epsilon}{u_*^3} \tag{8}$$

where $\epsilon$ is the turbulent kinetic energy dissipation rate, which is modelled herein as (Kaimal and Finnigan, 1994)

$$\phi_\epsilon^{2/3} = \begin{cases} 1 + 0.5|\zeta|^{2/3}, & \zeta \leq 0 \\ (1 + 5\zeta)^{2/3}, & \zeta \geq 0 \end{cases} \tag{9}$$

Equations (6) and (7) lead to the following relationships

$$\frac{S_v}{S_u} \approx \frac{S_w}{S_u} \simeq 1.33 \text{ at } f_r \gg 1 \tag{10}$$

Equation (10) is known as the assumption of local isotropy in the inertial subrange (Kolmogorov, 1941), although the latter may be reached without local isotropy (Mestayer, 1982; Chamecki and Dias, 2004). Equations (6) and (7) are convenient relationships not only to assess the data quality (Peña et al., 2019; Cheynet et al., 2019), but also to study the influence of waves on atmospheric turbulence, since a deviation from the $4/3$ law may be observed in the case of mixed-sea or swell (Smedman
et al., 2003).

### 3.3 The coherence of turbulence

The coherence of turbulence describes the spatial correlation of eddies. The real part of the coherence called co-coherence, is one of the governing parameters for the structural design of wind turbines (IEC 61400-1, 2005). At vertical separations, the



co-coherence $\gamma_{ij}$, where $i = \{u, v, w\}$, is defined as:

$$\gamma_i(z_1, z_2, f) = \frac{\text{Re}\{S_i(z_1, z_2, f)\}}{\sqrt{S_i(z_1, f)S_i(z_2, f)}} \qquad (11)$$

where $S_i(z_1, z_2, f)$ is the two-point cross-spectral density between heights $z_1$ and $z_2$, whereas $S_i(z_1, f)$ and $S_i(z_2, f)$ are the one-point spectra estimated at heights $z_1$ and $z_2$, respectively.

Davenport (1961) proposed an empirical model to describe the co-coherence for vertical separations, which depends only on a decay parameter $c^i$ and a reduced frequency $n$:

$$\gamma_i(n) \approx \exp\left(c^i n\right) \qquad (12)$$

$$n = \frac{2fd_z}{\overline{u}(z_1) + \overline{u}(z_2)} \qquad (13)$$

where $d_z = |z_1 - z_2|$. For three heights $z_1 > z_2 > z_3$ such that $z_1 - z_2 = z_2 - z_3$, Davenport's model predicts that $\gamma_i(z_1, z_2, f)$ and $\gamma_i(z_2, z_3, f)$ collapse onto a single curve when expressed as a function of $n$. This behavior, referred to as the Davenport's similarity herein, is questioned by Bowen et al. (1983) for vertical separations and by Kristensen et al. (1981) and Sacré and Delaunay (1992) for lateral separations.

Bowen et al. (1983) modified the Davenport model by assuming that $c^i$ was a linear function of the distance, i.e.

$$c^i = c_1^i + \frac{2c_2^i d_z}{(z_1 + z_2)} \qquad (14)$$

Equation (14) reflects the blocking by the ground or the sea surface, which leads to an increase of the co-coherence with measurement height. This equation implies that the co-coherence decreases more slowly than predicted by the Davenport model if measurements are conducted far from the surface and at short separations. On the other hand, the co-coherence may decrease faster than predicted by the Davenport model if the measurements are associated with large separation distances. This implies that fitting the Davenport model to measurements with short or large separations may only lead to an inadequate design of wind turbines.

The model by Bowen et al. (1983) was further modified by Cheynet (2019) by including a third decay parameter $c_3^i$ to account for the fact that the co-coherence cannot reach values of 1 at zero frequency, unless the separation distance is zero. This led to the following three-parameter co-coherence functions, which is herein referred to as the modified Bowen model:

$$\gamma_{ii}(z_1, z_2, f) = \exp\left\{-\left[\frac{|z_2 - z_1|}{\overline{u}(z_1, z_2)}\sqrt{(c_1^i f)^2 + (c_3^i)^2}\right]\right\}$$
$$\times \exp\left(-\frac{2c_2^i f|z_2 - z_1|^2}{(z_1 + z_2)\overline{u}(z_1, z_2)}\right) \qquad (15)$$

It should be noted that both $c_1^i$ and $c_2^i$ are dimensionless whereas $c_3^i$ has the dimension of the inverse of a time. Following Kristensen and Jensen (1979), $c_3^i \propto 1/T$ where $T$ is a time scale of turbulence. Therefore, low values of $c_3^i$ are associated with a co-coherence converging toward 1 at low frequencies for which the separation distance is small compared to a typical turbulence





length scale. The rotor diameter of multi-megawatt OWTs commissioned after 2015 in the North Sea is slightly larger than $150\,\mathrm{m}$. For such structures, assuming $c_3^i \approx 0$ may no longer be appropriate.

IEC 61400-1 (2005) recommends the use of two empirical coherence formulations. The first one was derived based on the exponential coherence proposed by Davenport (1961), which read as

$$\gamma_u(f, d_z) = \exp\left\{-12\left[\sqrt{\left(\frac{fd_z}{\overline{u}_{hub}}\right)^2 + \left(0.12\frac{d_z}{8.1L_c}\right)^2}\right]\right\}$$
(16)

where $\overline{u}_{hub}$ is the mean wind speed at the hub height and

$$L_c = \begin{cases} 0.7z, & z \leq 60\,\mathrm{m} \\ 42\,\mathrm{m}, & z \geq 60\,\mathrm{m} \end{cases}$$
(17)

The second coherence model was derived based on a spectral tensor of homogeneous turbulence (Mann, 1994) but is not described in detail here. Further assessments of this model can be found in Mann (e.g. 1994), Saranyasoontorn et al. (2004) or
Cheynet (2019).

## 4 Data processing

Sonic anemometer data monitored continuously from May 1994 to July 1995 were selected. No data was collected in July 1994 and October 1994, leading to 13 months of available records. The sonic anemometer at $z = 18\,\mathrm{m}$ was chosen as the reference sensor throughout the data processing because the measurements at $z = 45\,\mathrm{m}$ were associated with a low signal-to-noise ratio, which prevented a reliable estimation of the Obukhov length at this height. On the other hand, the measurements at $z = 6\,\mathrm{m}$
were suspected to be located during a substantial amount of time in the wave boundary layer (WBL) (Sjöblom and Smedman, 2003a). This layer is also called as the wave sublayer by Emeis and Türk (2009), who suggest that its depth is approximately $5H_s$, although there is no consensus on the depth of the WBL. The WBL is defined hereinafter in a similar fashion as by Edson and Fairall (1998), i.e. it is the layer above the sea surface where $\phi_m(\zeta)$ or $\phi_w(\zeta)$ deviate from MOST. In this regard, the present definition differs slightly from Edson and Fairall (1998) or Sjöblom and Smedman (2003a) who did not study $\phi_w(\zeta)$ above the
sea surface.

Both the double rotation technique and the sectoral planar fit (PF) method (Wilczak et al., 2001) were considered to correct the tilt angles of the SAs. The choice of the algorithm relied on a comparison between the friction velocity $u_*$ estimated using eq. (2) and the method by Klipp (2018), which does not require any tilt correction. The latter method provides an estimate
$u_{*R}$ of the friction velocity using the eigenvalues of the Reynolds stress tensor. Following this comparison, the double rotation technique was found to provide, in the present case, slightly more reliable results than the PF algorithm (see section 5.3). It should be noted that this finding is likely specific to the Vindeby dataset as the planar fit method usually provides better estimates of the covariance of turbulence.

The time series were sometimes affected by the outliers, which were removed using a moving median window based on $5\,\mathrm{min}$
window length. The local median values were then used to compute the median absolute deviation (MAD) (Leys et al., 2013).





Data located more than five MAD away from the median were classified as outliers and replaced with NaNs. The same outlier detection algorithm was also used for the sea surface elevation data, but with a moving window of $180\,\mathrm{s}$.

The moving average and a moving standard deviation with a window length of $10\,\mathrm{min}$ were used to assess the first- and second-order stationarity of the velocity data, respectively. The time series were considered as stationary when the two following criteria were fulfilled: (1) the maximum absolute relative difference between the moving mean and the static mean was lower than a threshold value of 20%; (2) for the moving standard deviation, the maximum absolute relative difference was also used with a threshold value of 40%. The choice of a larger threshold value for the moving standard deviation test is justified by the larger statistical uncertainty associated with the variance of a random process compared to its mean (Lumley and Panofsky, 1964).

Velocity records with an absolute value of skewness larger than 2 or a kurtosis below 1 or above 8 are likely to display an unphysical behaviour due to e.g. high measurement noise (Vickers and Mahrt, 1997) and were subsequently dismissed. The statistical uncertainties of the records were quantified as by Wyngaard (1973) and Stiperski and Rotach (2016):

$$a_{ii}^2 = \frac{4z}{T\overline{u}}\left[\frac{\overline{i'^4}}{\sigma_i^4} - 1\right] \tag{18}$$

$$a_{uw}^2 = \frac{z}{T\overline{u}}\left[\frac{\overline{(u'w')^2}}{u_*^4} - 1\right] \tag{19}$$

$$a_{vw}^2 = \frac{z}{T\overline{u}}\left[\frac{\overline{(v'w')^2}}{u_*^4} - 1\right] \tag{20}$$

where $a_{ij}$ with $i,j = (u,v,w)$ is the uncertainty associated with the variance and covariance estimates. Time series with a large random error, i.e. $a_{ii} > 0.20$ or $a_{ij} > 0.50$, $i \neq j$, were excluded.

The records with a mean wind speed below $5.0\,\mathrm{m\,s^{-1}}$ at $18\,\mathrm{m}$ amsl were discarded. Assuming a logarithmic mean wind profile, a near-neutral atmosphere, and a roughness length $z_0 = 2 \times 10^{-4}\,\mathrm{m}$, the corresponding mean wind speed at $18\,\mathrm{m}$ amsl is $5.7\,\mathrm{m\,s^{-1}}$. The present choice of a lower threshold mean wind speed is, therefore, consistent with the cut-in wind speed of large offshore wind turbines, which is $5.0\,\mathrm{m\,s^{-1}}$ at hub height. It ensures also a consistent comparison of the spectral characteristics of turbulence with the data collected at FINO1, where the lowest mean wind speed considered was $5.0\,\mathrm{m\,s^{-1}}$ at $80\,\mathrm{m}$ amsl.

The power spectral density (PSD) estimates of the velocity fluctuations were evaluated using Welch's method (Welch, 1967) with a Hamming window, three segments, and 50% overlap. The spectra were ensemble-averaged using the median of multiple $30\,\mathrm{min}$ time series that passed the data-quality tests described above and were smoothed by using a bin-averaging over logarithmically-spaced bins. The co-coherence estimates were also computed using Welch's method but using eight segments and 50% overlap to further reduce the statistical uncertainty.

Table 1 displays the percentage of samples at each measurement height that failed the data-quality assessment. It relies on initial data availability of 86%, 97%, and 86% for the anemometers at $6\,\mathrm{m}$ amsl, $18\,\mathrm{m}$ amsl, and $45\,\mathrm{m}$ amsl, respectively. Following the criteria used in the data processing and Table 1, the percentage of data considered for the analysis was 69%, 76%, 45% at $6\,\mathrm{m}$, $18\,\mathrm{m}$ and $45\,\mathrm{m}$ respectively. These percentages correspond to 1566 time series of $30\,\mathrm{min}$ duration for the SA at $6\,\mathrm{m}$, 1771 time series at $18\,\mathrm{m}$, and 854 at $45\,\mathrm{m}$. The data from SA at $45\,\mathrm{m}$ shows the highest portion of non-stationary and large



**Table 1.** Percentage of the records that failed the quality-data assessment.

|  | 6 m | 18 m | 45 m |
| --- | --- | --- | --- |
| NaNs $> 5\%$ | 5% | $< 1\%$ | 22% |
| Unphysical kurtosis and skewness | 4% | 3% | $< 1\%$ |
| Non-stationary | 9% | 15% | 19% |
| Large statistical uncertainties | 2% | 4% | 22% |

statistical uncertainties compared to the other SAs. Furthermore, the SA at $45\,\mathrm{m}$ also contained the highest fraction of NaN in the time series, which testified due to a large number of outliers. The larger fraction of data removed for the anemometer at $45\,\mathrm{m}$
is attributed to the observed uncorrelated white noises in the signal. This measurement noise, which may be linked to the length of the cable joining the anemometer and the acquisition system, is usually small for wind speed above $10\,\mathrm{m\,s^{-1}}$. Therefore, it was decided not to filter it out using digital low-pass filtering techniques. Time series that were flagged as non-physical made up only $< 5\%$ for each SA in the present datasets, likely because the test was applied after the outlier detection algorithm. The portion of non-stationary time series increased with height (see Table 1). Closer to the surface, the eddies are smaller and are
less likely to be affected by the sub-meso or mesospheric atmospheric motion, which contribute to non-stationary fluctuations (Högström et al., 2002).

## 5  Results

### 5.1  Applicability of MOST

In the atmospheric surface layer, the friction velocity $u_*$ is often assumed constant with the height (constant flux layer). However,
fig. 4 shows that the friction velocity is generally larger at $6\,\mathrm{m}$ than at the other two measurement heights, especially under stable conditions. The larger value of $u_*$ at $6\,\mathrm{m}$ amsl than at $18\,\mathrm{m}$ amsl may reflect the contribution of the wave-induced stress to the total turbulent stress in the few meters above the sea surface (Janssen, 1989; Tamura et al., 2018).

The applicability of MOST is assessed by studying $\phi_w$ and $\phi_m$ as a function of $\zeta$. Each sub-panel of fig. 5 shows $\phi_w$ at a different measurement height, whereas the black solid line corresponds to eq. (5). Under near-neutral conditions ($|\zeta| \leq 0.1$),
$\phi_w \approx 1.32$ at $45\,\mathrm{m}$ amsl, $\phi_w \approx 1.24$ at $18\,\mathrm{m}$ amsl, and $\phi_w \approx 1.17$ at $6\,\mathrm{m}$ amsl. In flat and uniform terrains, a ratio of $\phi_w \approx 1.25$ is generally found for near-neutral conditions (Panofsky and Dutton, 1984, Table 7.1). The error bars associated with the estimates at $45\,\mathrm{m}$ are likely related to the presence of the uncorrelated white noises in the velocity records (Kaimal and Finnigan, 1994, section 7.4.2), which leads to an underestimation of $u_*$. The wave-induced stress may increase the friction velocity at $6\,\mathrm{m}$ amsl, and therefore, a lower-than-expected $\phi_w$.
For an unstable atmosphere ($\zeta < 0$), the values of $\phi_w$ estimated at $18\,\mathrm{m}$ remain under eq. (5), which were not observed on FINO1 (Cheynet et al., 2018). It is unclear whether the lower-than-expected value of $\phi_w$ is due to the contribution of wave-induced stress to $u_*$ or an underestimation of $w'$ due to probe-induced flow-distortion. At $18\,\mathrm{m}$ amsl, the ratio $S_w / S_u$





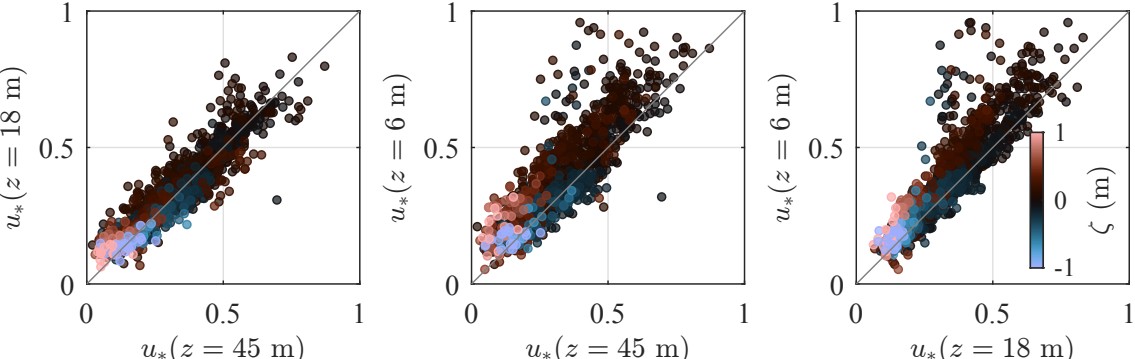

**Figure 4.** Friction velocity estimated by the three sonic anemometers on SMW for a wide range of stability conditions with $|\zeta| < 2$.

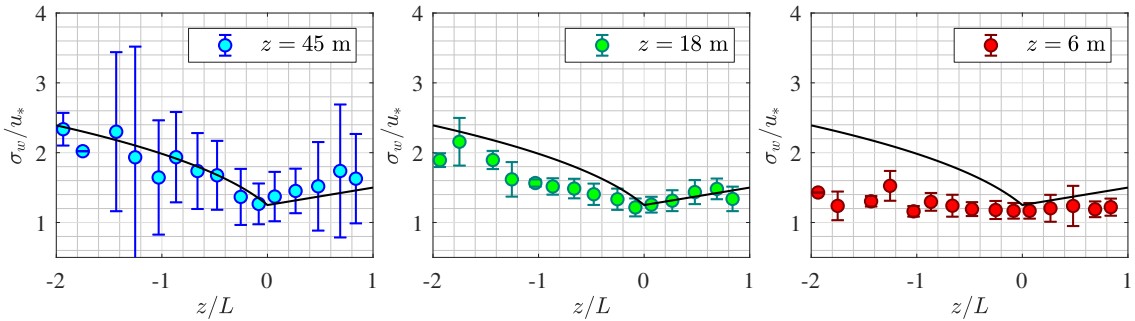

**Figure 5.** Variation of the $\phi_w = \sigma_w/u_*$ with the non-dimensional Obukhov length $\zeta$ estimated from SA at $18\,\text{m}$ amsl, superposed with the empirical value (black line) provided by Kaimal and Finnigan (1994). The error bar represents the interquartile range.

converges toward 1.2 in near-neutral conditions, i.e. slightly below the theoretical value of 1.33 (Kolmogorov, 1941). However, this value is similar to the ratio estimated from data measured at FINO1 platform. Therefore, the sensor-induced flow distortion is unlikely to explain the deviation between eq. (5) and the estimated values at $18\,\text{m}$ amsl. At $6\,\text{m}$ amsl, the values of $\phi_w$ are fairly constant because the local estimate of $\zeta$ shows a great portion of near-neutral conditions than at $18\,\text{m}$ amsl.

The similarity relations describing the mean wind speed profile agrees well with the sonic anemometer measurements under all stability conditions except between the sensor at $6\,\text{m}$ and $18\,\text{m}$ amsl at $\zeta > 0.3$. In fig. 5, the friction velocity is averaged between the two heights selected. Therefore, the observed deviation may be partly due to the contribution of the wave-induced stress to the friction velocity. The right panel of fig. 6 does not show such deviation, maybe because the friction velocity estimated at $45\,\text{m}$ amsl is slightly underestimated due to the high-frequency noises in the velocity records of the top sensor. The turbulence measurement collected at $6\,\text{m}$ amsl appears to be affected during a substantial amount of the time by the waves, which leads to clear deviations from MOST. This further justifies the use of the sonic anemometer at $18\,\text{m}$ amsl to estimate the non-dimensional Obukhov length $\zeta$.





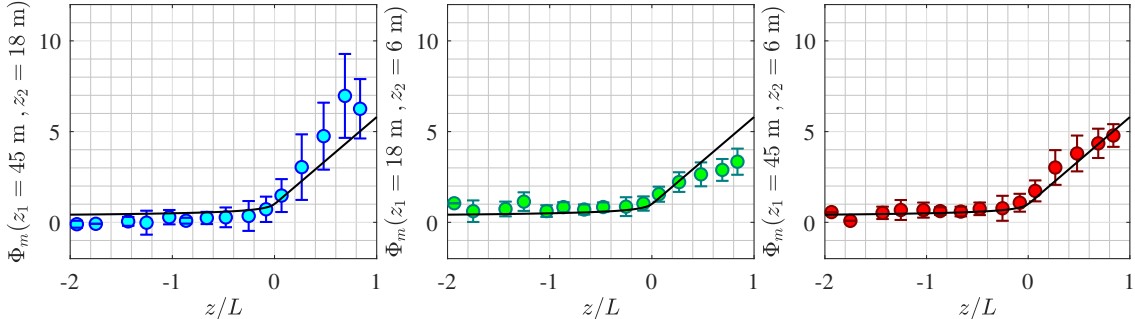

**Figure 6.** Variation of $\phi_m$ with the non-dimensional Obukhov length $\zeta$ estimated from SA at $18\,\mathrm{m}$ amsl. The solid black line is eq. (4) and the error bar represents the interquartile range.

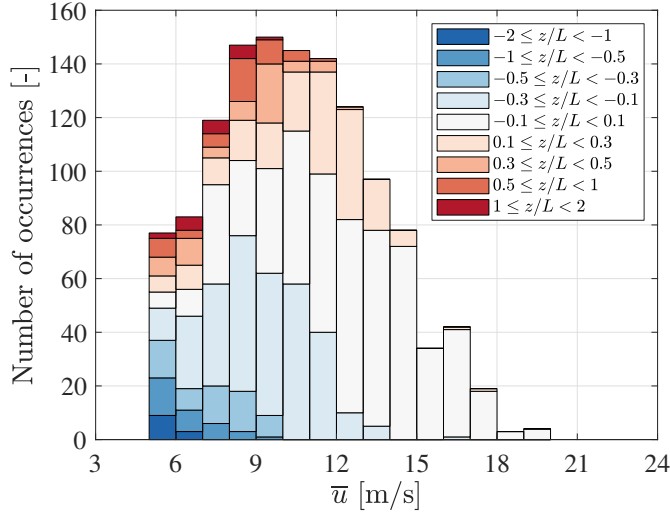

**Figure 7.** Stability distribution as a function of mean wind speed for the considered fetch (220°-330°) at height $z = 18\,\mathrm{m}$.

A better agreement between the estimates and the empirical values of $\phi_m$ and $\phi_w$ is obtained if the friction velocity at $18\,\mathrm{m}$ amsl is used instead of the local values. Even when doing so, the estimated values of $\phi_w$ at $6\,\mathrm{m}$ amsl deviate from MOST. The validity of MOST in the vicinity of the sea surface under wind-sea conditions is, therefore, more disputed in the present case than in previous studies (e.g. Drennan et al., 1999).

The distribution of $\zeta$ as a function of the mean wind speed $\overline{u}$ is given in fig. 7 for the sector between 220° and 330°. The majority (82%) of the stationary records samples were associated with a wind speed between $7\,\mathrm{m\,s^{-1}}$ to $15\,\mathrm{m\,s^{-1}}$ at $18\,\mathrm{m}$ amsl. Non-neutral conditions are defined herein as situations where $|\zeta| > 0.1$. They represent 69% of the samples at $\overline{u} < 12\mathrm{m\,s^{-1}}$ and 12% at $\overline{u} \geq 12\mathrm{m\,s^{-1}}$. The distribution of the atmospheric stability conditions is in overall agreement with Barthelmie (1999) and Sathe and Bierbooms (2007) for Vindeby site.





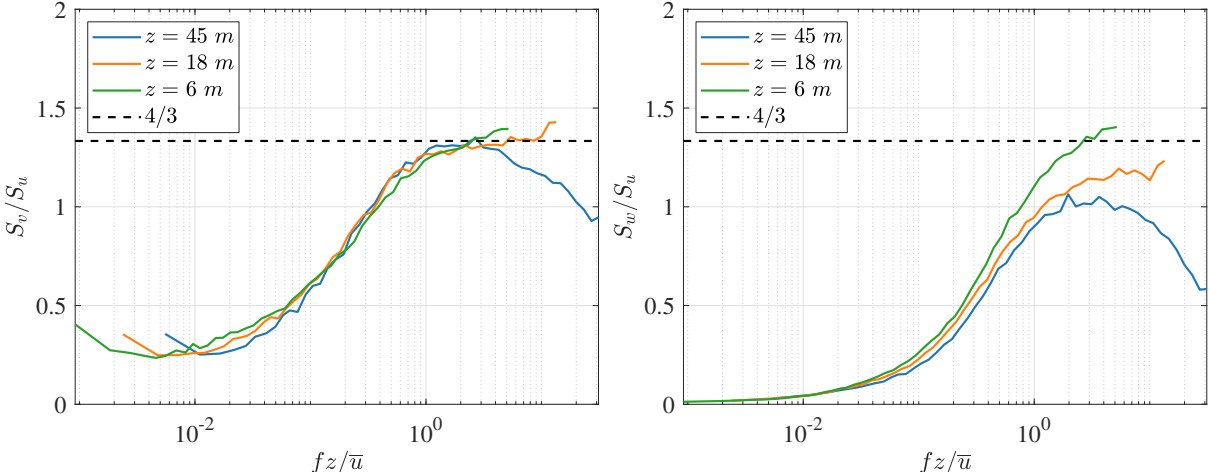

**Figure 8.** Spectral ratio $S_v/S_u$ (left panel) and $S_w/S_u$ (right panel) for near-neutral conditions.

## 5.2 Local isotropy

The spectral ratios $S_v/S_u$ and $S_w/S_u$ for near-neutral conditions ($-0.1 \leq \zeta \leq 0.1$) are presented in fig. 8. As documented by e.g. Chamecki and Dias (2004), Cheynet et al. (2018) or Peña et al. (2019), the isotropic values of $S_v/S_u$ are reached more easily than by $S_w/S_u$. The presence of significant measurement noises in the velocity data at the top sensor leads to ratios $S_v/S_u$ and $S_w/S_u$ that reach a maximum at $f_r \approx 3$ before decreasing at higher frequencies. The isotropic values of $S_w/S_u$ are only reached for the sensors located at $6\,\mathrm{m}$ amsl. Smedman et al. (2003) observed that the maximum value of $S_w/S_u$ is close

to or below unity in the presence of a swell sea-state. Figure 8 shows that a similar deduction is not applicable here, because the wind-sea conditions are predominant. At $18\,\mathrm{m}$ amsl, the maximum value of the ratio $S_w/S_u$ is similar to those observed in Cheynet et al. (2018) or Peña et al. (2019) and can be attributed to flow distortion by the instrument. It is quite remarkable that the isotropic value of $S_w/S_u$ is reached by the sensor closest to the surface, where flow distortion by the tower structure is usually larger than at the upper levels.

## 5.3 Estimation of the friction velocity


Figure 9 compares the friction velocity estimates $u_{*R}$ by Klipp (2018) and $u_*$ when applying the double rotation of the anemometer axes for various atmospheric stratifications. In general, the resulting friction velocity from both methods is in good agreement. The average correlation coefficient for all heights is 0.985 for $|\zeta| \leq 2$. The PF algorithm leads to a slightly larger scatter between $u_{*R}$ and $u_*$, where the average correlation coefficient from all heights is 0.976 for $|\zeta| \leq 2$ (Table 2). The double

rotation algorithm seems to give a smaller deviation between $u_{*R}$ and $u_*$ than the PF algorithm in the present study, which justified the adoption of the double rotation as tilt correction method herein.





**Table 2.** Correlation coefficients between $u_{*R}$ and $u_*$ using the planar fit (PF) or double rotation (DR)

|  | $|z/L| \leq 0.1$ | | | $0.1 < |z/L| \leq 2.0$ | | |
|---|---|---|---|---|---|---|
|  | 6 m | 18 m | 45 m | 6 m | 18 m | 45 m |
| PF | 0.989 | 0.976 | 0.962 | 0.981 | 0.954 | 0.942 |
| DR | 0.995 | 0.986 | 0.973 | 0.989 | 0.968 | 0.963 |

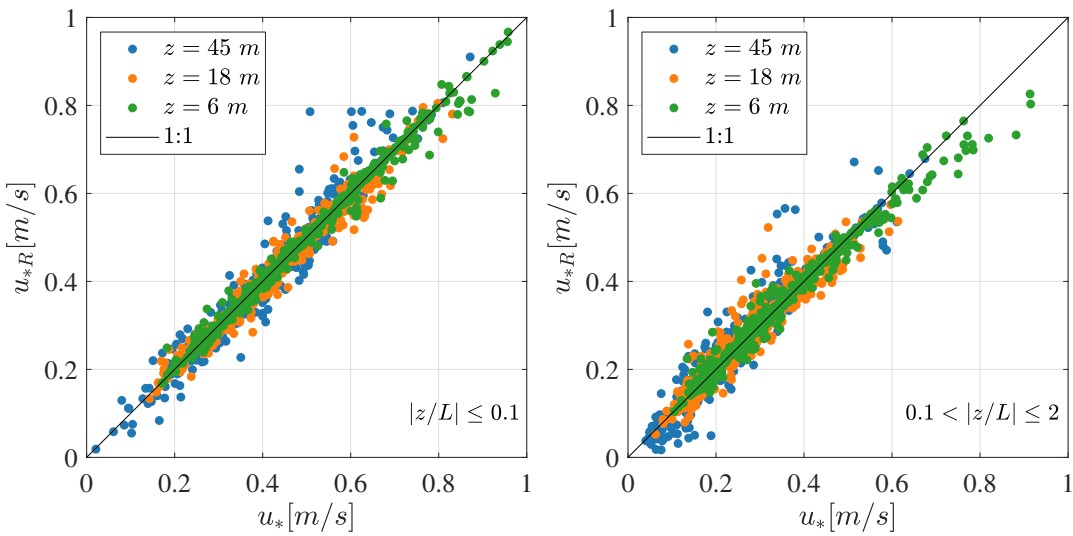

**Figure 9.** Friction velocity computed using the eddy covariance method with the double rotation method compared with the Klipp method. The left panel considers only $|z/L| \leq 0.1$ and the right panel considers $0.1 < |z/L| \leq 2$.

Klipp (2018) noted that $u_{*R}$ is appropriate to estimate the friction velocity if the thermal stratification of the atmosphere is neutral only. Yet, fig. 9 suggests that Klipp's method is performing well for non-neutral conditions too, as highlighted by the correlation coefficients in Table 2, which vary between 0.963 to 0.989. Additional studies using measurements from other coastal or offshore sites are needed to assess if such observations are recurring.

The angle between the stress vector and the wind vector is given as $\alpha = \arctan\left(\overline{v'w'}/\overline{u'w'}\right)$ (Grachev et al., 2003). It is found that $\alpha$ increases from $8°$ at $6\,\mathrm{m}$ amsl to $13°$ at $45\,\mathrm{m}$ amsl when $\overline{v'w'} < 0$. When $\overline{v'w'} > 0$, $\alpha$ is almost constant with the height with an average value of $-7°$. The relatively low value of $\alpha$ therefore suggests that the direction of the wind-wave-induced stress is fairly well aligned with the mean wind direction near SMW.

## 5.4 Wind-wave interactions

The unusual turbulence characteristics identified at $6\,\mathrm{m}$ amsl in section 5.1 are explored herein in terms of wind-wave interactions, using the wave elevation data collected by the acoustic wave recorder near SMW. A total of 925 high-quality samples collocated



in time with the wind velocity data studied herein were identified. Each wave elevation record was $30\,\mathrm{min}$ long and corresponded to a wind direction between $220°$ and $330°$.

The term "high-quality samples" refers to the sea surface elevation time histories $\eta(t)$ without flattened valleys or significant measurement noises at frequencies under $f_t = 0.10\,\mathrm{Hz}$, which were sometimes observed in the records. The identified wave peak period $T_p$ was generally located at frequencies above $0.20\,\mathrm{Hz}$, which justifies the choice of a threshold frequency of $0.10\,\mathrm{Hz}$. More precisely, the contribution of wave elevation data at frequencies below $f_t$ to the variance of the signal was negligible unless non-stationary fluctuations were recorded. The sea surface elevation skewness ranged from $-0.02$ to $0.37$ with a median

value of $0.17$, while the kurtosis varied from $2.7$ to $3.4$ and a median value of $3$. Therefore, $eta$ time series can be assumed Gaussian on average and the significant wave height $H_s$ was approximated as $4\sigma_\eta$ where $\sigma_\eta$ is the standard deviation of the sea surface elevation (Longwet-Higgins, 1952). Nonetheless, it should be emphasized that these results are concluded based on the measurement at one location near SMW, and the wave characteristics upstream of the mast are unexplored.

   Hourly hindcast data with a $2\,\mathrm{km}$ horizontal resolution (Tuomi and Huess, 2020) gives larger $H_s$ values than the measurement

data as shown in fig. 11. Close to the coast and in shallow water areas, the accuracy of hindcast data is usually lower. Also, the relatively low accuracy of the wave measurements leads to underestimated $H_s$ values. Nevertheless, the measured significant wave heights were below $1.5\,\mathrm{m}$ during 1994 and 1995 with a median value of $0.4\,\mathrm{m}$. The hindcast data provided $H_s$ values for wind-sea, primary and secondary swell. These data indicated that wind-sea conditions were largely predominant over swell conditions, as already mentioned in section 5.2.

The interactions between wind turbulence and the sea surface were explored in terms of the co-coherence and the quad-coherence between the vertical velocity component $w$ and the velocity of the wave surface $\dot{\eta} = \mathrm{d}\eta/\mathrm{d}t$. Similar approaches were adopted earlier by e.g. Grare et al. (2013) or Kondo et al. (1972) but using the squared coherence and without taking advantage of the ensemble average to reduce the systematic and random error, which are typically associated with the coherence function. In the present case, no clear coherence was found between $\dot{\eta}$ and $w$ for $H_s < 0.7\,\mathrm{m}$. For the sensor at $6\,\mathrm{m}$ amsl, a

non-zero coherence was discernible from the background noise at $0.7\,\mathrm{m} < H_s < 0.9\,\mathrm{m}$. The co-coherence and quad-coherence estimates were significantly different from zero for $0.9\,\mathrm{m} \leq H_s$, as illustrated in fig. 10, where the ensemble averaging of the 60 samples was applied to reduce the random error. The inset in fig. 10 shows that the selected records are characterized by a single spectral peak $f_p$ located at frequencies between $0.20\,\mathrm{Hz}$ and $0.25\,\mathrm{Hz}$, which is the frequency range where the quad-coherence is substantially different from zero. The observed behaviour of the co-coherence and the quad-coherence at this frequency range

may show the $90°$ out-of-phase fluctuations between $\dot{\eta}$ and $w$, where the latter is lagging. The co-coherence and quad-coherence estimates between $\dot{\eta}$ and the horizontal wind component $u$ were also investigated but were nearly zero for the three sonic anemometers on SMW.

   It should be noted that the influence of the sea surface elevation on the vertical turbulence was not clearly visible in the one-point vertical velocity spectra $S_w$, except for $H_s > 1.2\,\mathrm{m}$, where a weak spectral peak near $0.2\,\mathrm{Hz}$ was distinguishable. The

wave-induced wind component is generally much less compared to the wind turbulence (Weiler and Burling, 1967; Kondo et al., 1972; Naito, 1983). An exception is the case of weak wind and swell conditions which are more likely to result in the observation of a sharp spectral peak near $f_p$ in the $S_w$ spectrum. Nonetheless, as previously mentioned, such conditions are rare near SMW.



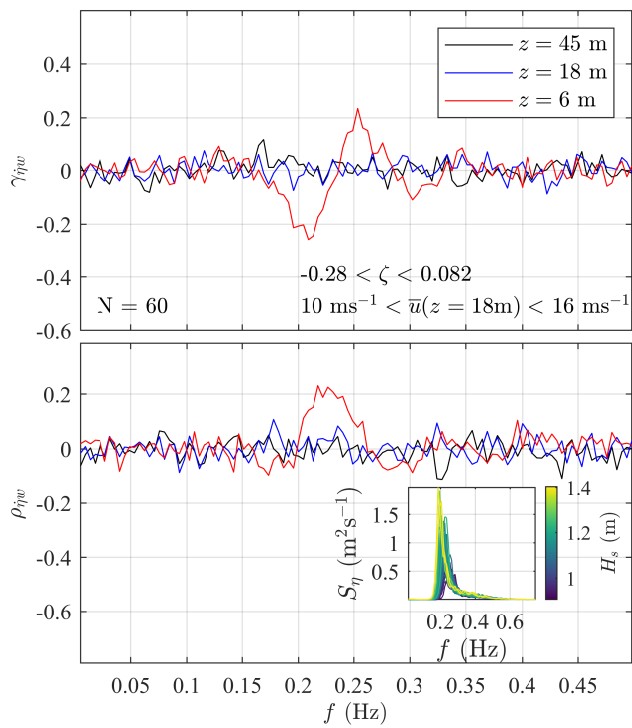

**Figure 10.** Co-coherence $\gamma_{\dot{\eta}w}$ and quad-coherence $\rho_{\dot{\eta}w}$ between the velocity of the wave surface $\dot{\eta}$ and the vertical wind velocity $w$ from the three sonic anemometers on SMW. The inset shows the individual wave elevation spectra $S_\eta$ associated with $H_s > 0.9\,\text{m}$ (60 samples) used to estimate $\gamma_{\dot{\eta}w}$ and $\rho_{\dot{\eta}w}$.

According to Grare et al. (2013), the contribution of wave-induced momentum flux to the total momentum flux $\overline{u'w'}$ is positive for relatively young waves, i.e. $C_p/u_* < 40$, where $C_p$ is the phase speed at the wave spectral peak. Using shallow water theory,

the wave age near SMW is $C_p/u_* < 30$ most of the time, which would partly explain the larger friction velocity measured at $6\,\text{m}$ amsl compared to $18\,\text{m}$ amsl. It should be noted, however, that the study by Grare et al. (2013) was conducted in deep waters with measurements located at heights lower than $5H_s$. They noted that the positive contribution of the wave-induced momentum flux they measured was close to or below 10% of the total momentum flux. In the present case, the sonic anemometers were located at heights close to or larger than $5H_s$ most of the time. Nevertheless, the momentum flux $\overline{u'w'}$ estimated at $6\,\text{m}$ amsl

was, on average, 21% and 18% larger than those at $45\,\text{m}$ and $18\,\text{m}$, respectively. For stable conditions with $\zeta > 0.2$, $\overline{u'w'}$ at $6\,\text{m}$ was 50% larger than at $45\,\text{m}$ amsl and $18\,\text{m}$ amsl. Both results displayed in section 5.3 and fig. 10 suggest that the wave sublayer, as defined by Emeis and Türk (2009), may be deeper than $5H_s$ near SMW.

The limited number of data showing a clear correlation between the velocity of the sea surface and the vertical wind component may imply that the deviations from MOST observed at $6\,\text{m}$ amsl may also be influenced by the heterogeneous surface roughness

nearby SMW. For the wind sectors selected, an increase in the average $H_s$ toward SMW can be seen in fig. 11. This would result





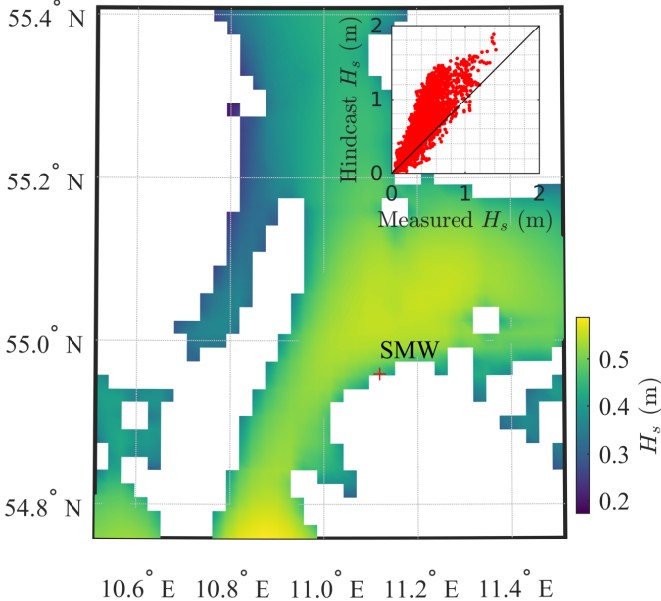

**Figure 11.** Median value of the hourly significant wave height for the year 1995 near SMW provided by the hindcast data (Tuomi and Huess, 2020). The inset compares the measured and modelled $H_s$ values between 1994 and 1995.

in spatially-varying surface roughness between the upstream region and nearby SMW. In shallow water close to SMW (up to $300\,\mathrm{m}$), it is likely to observe the presence of non-linear wave steepness that would contribute to enhanced surface roughness and thus larger turbulent stresses in the proximity of the mast. The variability of the surface roughness may be small enough so that $\phi_m$ follows MOST at $6\,\mathrm{m}$ amsl but not $\phi_w$. The latter is based on local measurements only and is, therefore, more sensitive

to a height-dependant friction velocity than $\phi_m$.

### 5.5 Turbulence spectra

Figure 12, fig. 13, and fig. 14 depict the PSD estimates respectively for the along-wind, cross-wind, and vertical velocity components as a function of the reduced frequency $f_r = fz/\overline{u}$ for nine stability classes. Surface-layer scaling is adopted, i.e. the PSDs are normalized with $u_*$ (eq. (2)) and $\phi_\epsilon^{2/3}$ (eq. (8)). Strongly non-neutral cases, defined as $|\zeta| > 2$ are not studied herein as

they are fairly uncommon for the dataset selected. The number of available samples for each stability class is denoted as $N$ and displayed in each sub-panel.

These three figures compare the estimated spectra at $z = 45\,\mathrm{m}$, $z = 18\,\mathrm{m}$, and $z = 6\,\mathrm{m}$ amsl with the empirical model established on the FINO1 platform (black solid line) at $z = 41.5\,\mathrm{m}$ amsl (Cheynet et al., 2018). The red curves represent the high-frequency asymptotic behaviour of surface-layer spectra for each stability class. It should be noted that the latter curves do

not indicate when the inertial subrange starts since the frequencies they cover were arbitrarily chosen.





In fig. 12, the maximum values of the normalized spectra for near-neutral conditions are close to unity, as described by Kaimal et al. (1972) which is another indication that the friction velocity was estimated properly. As highlighted in section 5.3, the anemometer at $6\,\mathrm{m}$ amsl recorded a friction velocity significantly larger than at $18\,\mathrm{m}$ amsl and $45\,\mathrm{m}$ amsl, which introduces a deviation from eq. (6) when the surface-layer scaling is applied to the velocity spectra. As mentioned in section 5.4, no spectral

peak around the wave spectral peak $f_p$ is visible in the $S_w$ spectrum, as expected, since ensemble averaging is applied and that such events were hardly observed at Vindeby.

As sketched in fig. 12, the velocity spectra estimated at $45\,\mathrm{m}$ amsl show systematic deviations from MOST under near-neutral and stable conditions, likely due to observed aforementioned uncorrelated high-frequency noises, which lead to an underestimation of the friction velocity. Under light and moderate unstable conditions, i.e. $-0.3 \leq \zeta \leq -0.1$, the velocity spectra

at $6\,\mathrm{m}$ and $18\,\mathrm{m}$ amsl are similar, which supports the idea that the wave sublayer is shallower than $6\,\mathrm{m}$. If $\zeta \leq -1$, deviations from MOST are clearer at both $45\,\mathrm{m}$ amsl and at $6\,\mathrm{m}$ amsl, which is also visible in fig. 14.

Following MOST, the normalized spectra at different heights should collapse onto one single curve at high frequencies, which was observed at heights between $40\,\mathrm{m}$ amsl and $80\,\mathrm{m}$ amsl at the FINO1 platform for $|\zeta| < 1$. However, this is not always the case in fig. 12, fig. 13, and fig. 14. Deviations from MOST for the PSDs estimates at $6\,\mathrm{m}$ amsl were expected due to the

contribution of wave-induced momentum flux (section 5.1). Regarding the velocity data at $45\,\mathrm{m}$ amsl, the measurement noises lift the high-frequency range of the velocity spectra above the spectral slope predicted by eq. (6) or eq. (7). At $18\,\mathrm{m}$ amsl, eqs. (6) and (7) predict remarkably well the velocity spectra at $f_r > 3$, indicating that surface-layer scaling is applicable at this height.

The presence of the spectral gap (Van der Hoven, 1957), separating the microscale fluctuations from the sub-meso and mesoscale ones, is noticeable at $\zeta > 0.3$, in line with previous observations (Smedman-Högström and Högström, 1975; Cheynet

et al., 2018). Under stable conditions, the spectral gap seems to move toward lower frequencies as the height above the surface decreases. This contrasts with the observations from an onshore mast on Østerild (Denmark) by Larsén et al. (2018), which indicated that the location of the spectral gap on the frequency axis was relatively constant with height.

Following Vickers and Mahrt (2003) the spectral gap timescale can be only a few minutes long under stable conditions. For $\zeta > 0.5$, the averaging period selected in the present study may be too large to provide reliable integral turbulence characteristics.

However, filtering out the mesoscale motion may not be desirable for structural design purposes since operating wind turbines experience both turbulence and mesoscale fluctuations (Veers et al., 2019). In this regard, the use of spectral flow characteristics to parametrise the wind loading on OWTs is preferable.

Under near-neutral conditions, the sensors at $6\,\mathrm{m}$ amsl and $18\,\mathrm{m}$ amsl are likely located in the so-called eddy surface layer (Högström et al., 2002; Drobinski et al., 2004), where the sea surface blocks the flow and distorts eddies. This leads to a flat

spectral peak because the integral length scale cannot be easily estimated. Such a spectral behaviour has also been observed above the eddy surface layer (Drobinski et al., 2004; Mikkelsen et al., 2017) but its consequences on wind turbine loads are unclear.

Deviations from the surface-layer scaling at $\zeta > 0.5$ are mainly due to the contribution of wave-induced stress to the total turbulent shear stress. This contribution was found to be highest for stable conditions compared to neutral and unstable ones



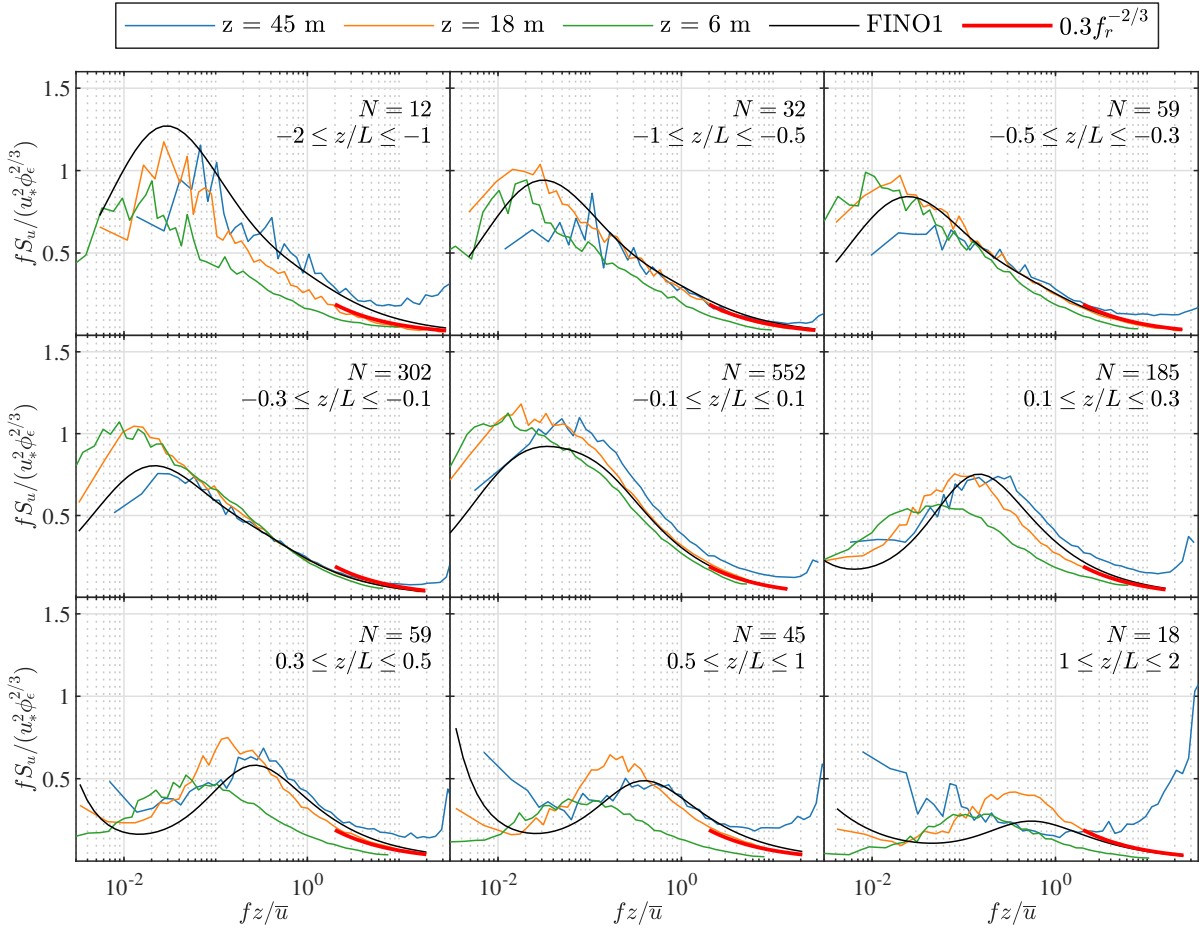

**Figure 12.** normalized spectra of the along-wind component at $45\,\mathrm{m}$, $18\,\mathrm{m}$ and $6\,\mathrm{m}$ amsl for various stability conditions. The red curve is derived from eq. (6) for reference but does not necessarily reflect the presence of an inertial subrange in the data.

(fig. 4). For $\zeta > 1$, further deviations from MOST may be related to the so-called local z-less stratification (Wyngaard and Coté, 1972) where turbulence is no longer scaled by the height above the ground.

Overall, the velocity spectra estimated at $18\,\mathrm{m}$ amsl and $45\,\mathrm{m}$ amsl at Vindeby match reasonably well with the empirical spectra estimated at $41\,\mathrm{m}$ amsl on the FINO1 platform for $-2 \geq \zeta < 1$. This comparison is encouraging for further explorations of the surface-layer turbulence characteristics at coastal and offshore sites. Nonetheless, detailed wind measurements at altitudes

$z \geq 100\,\mathrm{m}$ are favoured to get a complete overview of the turbulence characteristics in the MABL that could be relevant for OWT designs.



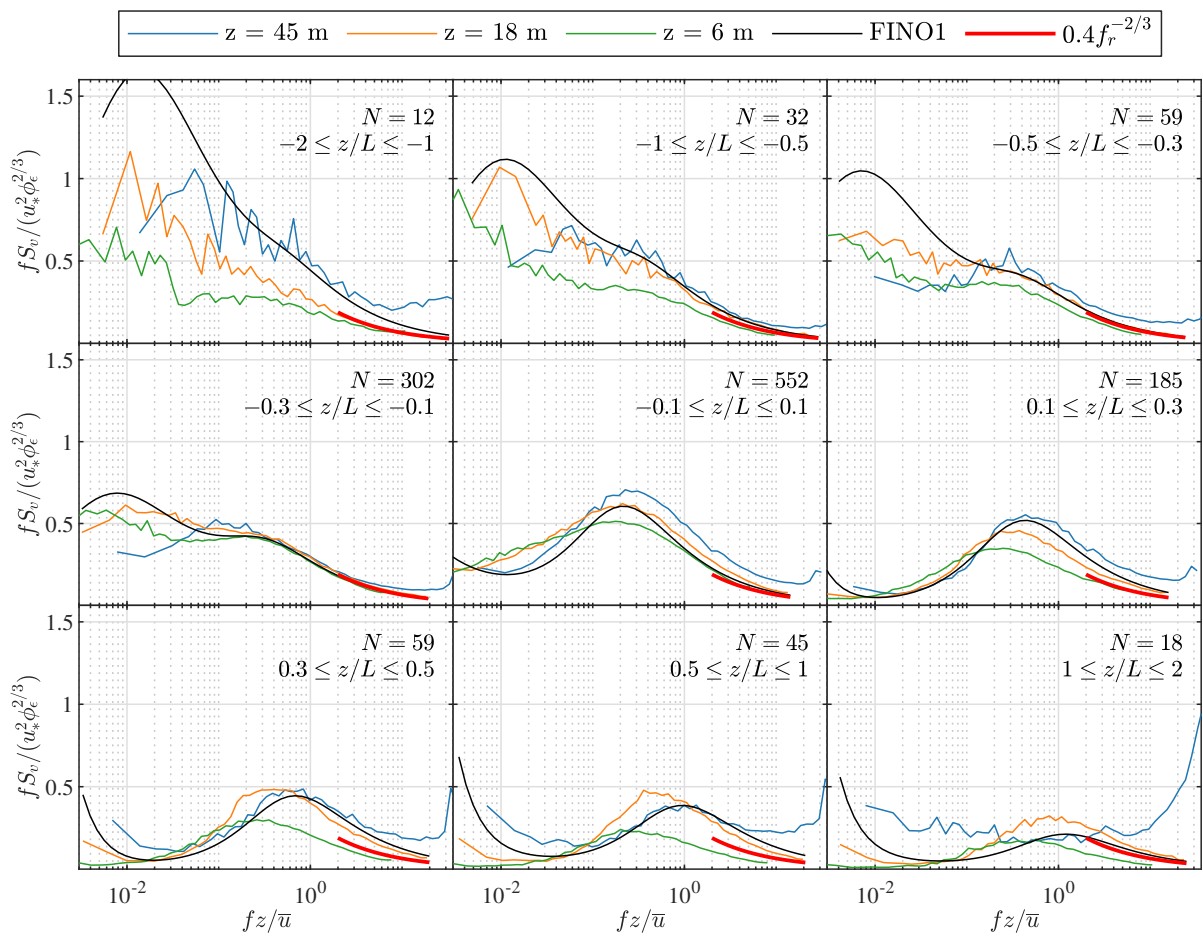

**Figure 13.** normalized spectra of the cross-wind component at $45\,\mathrm{m}$, $18\,\mathrm{m}$ and $6\,\mathrm{m}$ amsl for various stability conditions. The red curve is derived from eq. (7) for reference but does not necessarily reflect the presence of an inertial subrange in the data.

### 5.6 Co-coherence of turbulence

The vertical co-coherence of the along-wind, cross-wind, and vertical wind components are denoted as $\gamma_u$, $\gamma_v$, and $\gamma_w$, respectively. Under near-neutral conditions ($|\zeta| \leq 0.1$), these are expressed as a function of $kd_z$ in fig. 15 where $k = 2\pi f/\overline{u}$

using the assumption of frozen turbulence (Taylor, 1938). The co-coherence estimates are presented for three separation distances $d_z$ as three measurement heights ($z_1 = 45\,\mathrm{m}$, $z_2 = 18\,\mathrm{m}$, and $z_3 = 6\,\mathrm{m}$) were used. The co-coherence estimates collected on SMW are compared to the IEC coherence model (eq. (16)) and the modified Bowen model (eq. (15)). For the latter model, the parameters estimated on the FINO1 platform (Cheynet, 2019) are directly used since we aim to assess how similar are the turbulence characteristics of the atmosphere at FINO1 and Vindeby. The decay constants used for eq. (15) were therefore,

$[c_1^u, c_2^u, c_3^u] = [6.0, 17.8, 0.02]$ and $[c_1^w, c_2^w, c_3^w] = [2.7, 4.0, 0.16]$ as well as $[c_1^v, c_2^v, c_3^v] = [0, 23, 0.09]$.

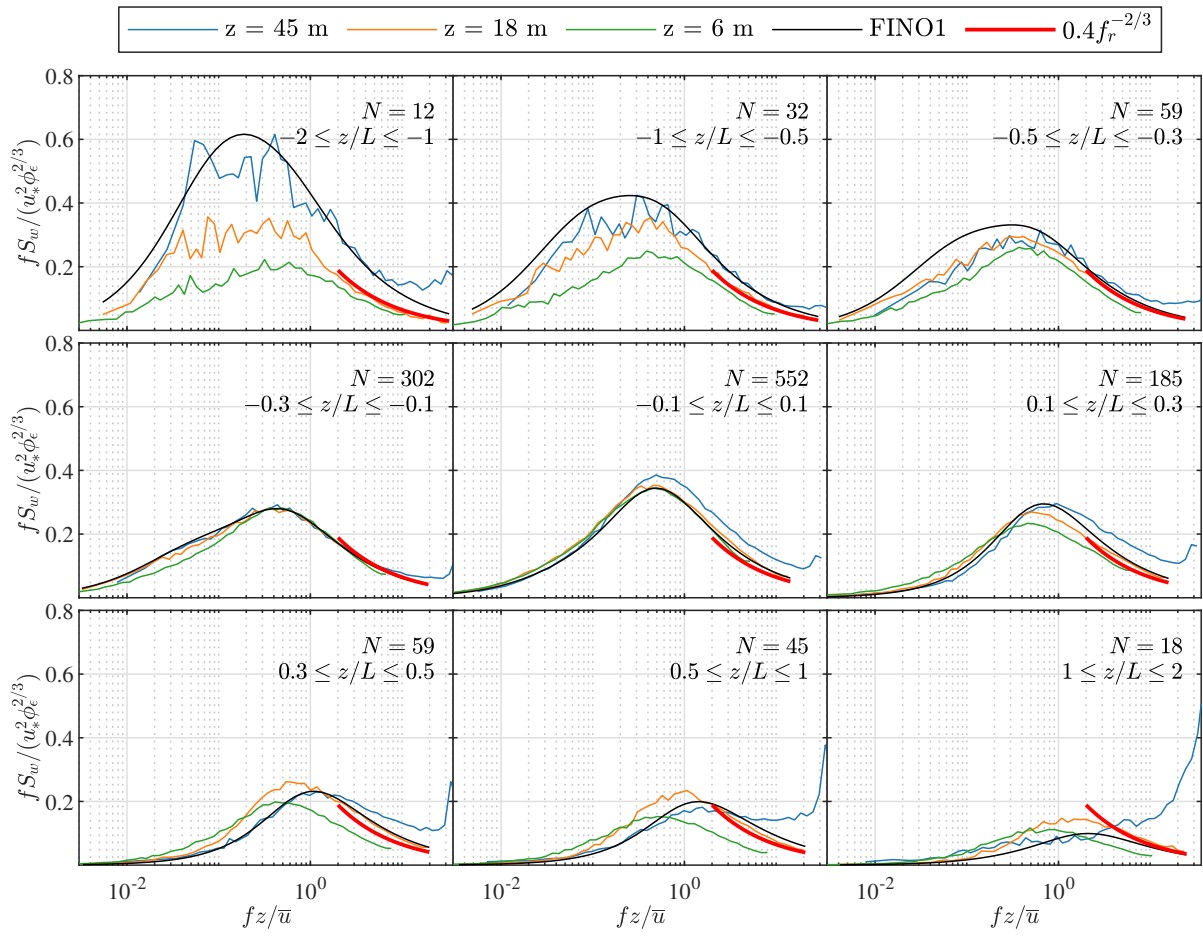

**Figure 14.** normalized spectra of the vertical wind component at $45\,\mathrm{m}$, $18\,\mathrm{m}$ and $6\,\mathrm{m}$ amsl for various stability conditions. The red curve is derived from eq. (7) for reference but does not necessarily reflect the presence of an inertial subrange in the data.

Figure 15 shows that the coefficients of the modified Bowen model estimated on the FINO1 platform apply exceptionally well to $\gamma_u$ estimated on SMW. Larger deviations are observed for the cross-wind components, for which $\gamma_v$ displays large negative values, especially for separations between $6\,\mathrm{m}$ amsl and $45\,\mathrm{m}$ amsl. On the FINO1 platform, the negative part of $\gamma_v$ was relatively small, which justified the use of eq. (15) with no negative co-coherence values. Following Bowen et al. (1983); ESDU 85020 (2002) or Chougule et al. (2012), the negative part is a consequence of the phase difference, and is non-negligible for the cross-wind component, which is also observed in the present case. Since this phase difference increases with the mean wind shear, it is more visible at SMW than at FINO1, where the measurements are at higher altitudes than at SMW.

The IEC exponential coherence model over-predicts $\gamma_u$ when the measurement height decreases and when the separation distance increases because this model follows fairly well Davenport's similarity, except at $kd_z < 0.1$. In Cheynet (2019), the Davenport model was suspected to lead to an overestimation of the turbulent wind loading on OWTs. The present results indicate

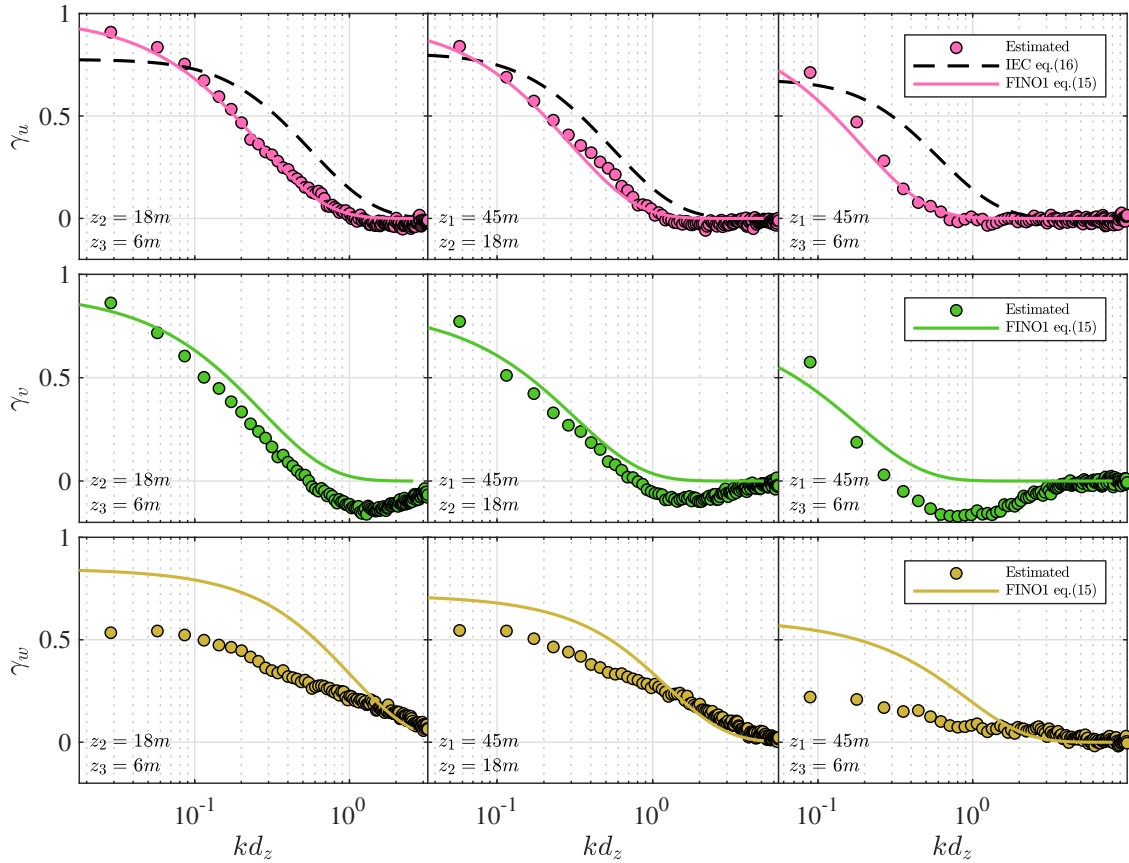

**Figure 15.** Co-coherences of the along-wind (top panels), lateral (middle panels), and vertical velocity components (lower panels) for $|\zeta| \leq 0.1$ at three different $d_z$. The dots represent the measurement and the lines mark the empirical values computed using the IEC exponential coherence (dashed line) and the Modified Bowen model (solid line) with the fitted coefficients from FINO1 (Cheynet, 2019).

that a similar overestimation may be obtained if the IEC exponential coherence model is used. Further studies are, however, needed to better quantify this possible overestimation since the lateral co-coherence is also required but could not be obtained at FINO1 nor SMW. The co-coherence for lateral separations may be obtained using Doppler wind lidar instruments (Cheynet et al., 2016, 2021) or wind sensors mounted on unmanned aerial vehicles (Wetz et al., 2021; Vasiljević et al., 2020) since

deploying multiple masts at offshore is likely too costly.

Figure 16 shows a clear variation of the estimated co-coherence with thermal stratification of the atmosphere for the three turbulence components. As observed by e.g. Soucy et al. (1982) or Cheynet et al. (2018) and modelled by Chougule et al. (2018), the vertical co-coherence is generally highest for convective conditions and smallest for stable conditions. Such results reinforce the idea that modelling the turbulent loading on offshore wind turbine using a coherence model established for neutral conditions

may only be appropriate for the ultimate limit state design but not for the fatigue life design.

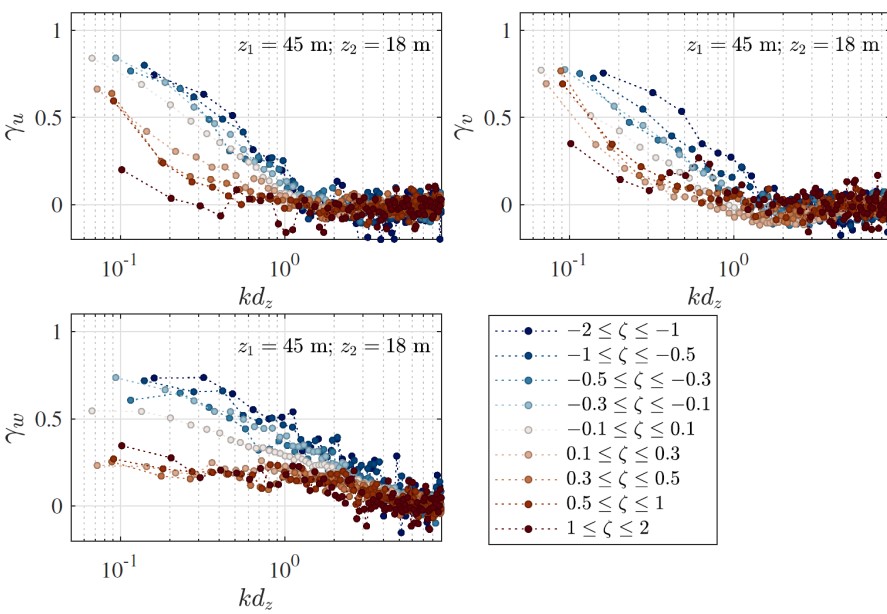

**Figure 16.** Co-coherences at separation between $z_1$ and $z_2$ for varying $z/L$. The upper left panel shows the estimated $\gamma_u$, the right upper panel displays the estimated $\gamma_v$, and the bottom left panel presents the estimated $\gamma_w$.

## 6 Relevancy of the database for load calculation of OWTs

While the present study provided a thorough overview of the MABL spectral turbulence characteristics with respect to the variation of the atmospheric stability at SMW, its direct applicability for the designs of OWTs has several limitations.

The first one is related to the fact that the presented results do not include the non-stationary conditions encountered in the field, which were removed before the analysis. About 20% of the data were disregarded as non-stationary to establish reliable spectra and co-coherence estimates. In the present case, non-stationary fluctuations were mainly associated with frequencies close to or below $0.05\,\mathrm{Hz}$. For typical spar-type and semisubmersible OWT floaters, these frequencies encompass the quasi-static motions and few lowest eigen-frequencies of the floaters (Jonkman and Musial, 2010; Robertson et al., 2014). Additionally, the non-stationary turbulence fluctuations could result in non-Gaussian loadings, which could further lead to underestimation of fatigue loading (Benasciutti and Tovo, 2006, 2007). Therefore, direct use of the presented results for floater motions and load estimations of OWTs may be associated with additional uncertainties, which could be addressed by introducing a safety factor.

Secondly, the present dataset was recorded at heights lower than the hub height of the recent and the future OWTs, which is around $130\,\mathrm{m}$ (e.g. GE Renewable Energy, 2021). At such heights, MOST may no longer be applicable (Peña and Gryning, 2008; Cheynet et al., 2021). Above the surface layer, the velocity spectra tend to be invariant with height, which is coarsely accounted for in IEC 61400-1 (2005) and suggested by preliminary observations from Doppler wind lidar instruments in coastal areas (Cheynet et al., 2021). Højstrup (1982) proposed an empirical turbulence spectral model for a convective boundary layer up to $0.5z_i$, where $z_i$ is the convective boundary layer height (Wyngaard and Coté, 1972), that could be used to characterize



turbulence above the surface layer. However, this model was not validated in the MABL, thus its applicability for OWT designs remains uncertain. For weakly stable atmosphere, turbulence can be parametrized using local similarity theory (Nieuwstadt,

1984; Sorbjan, 1986; Moraes, 1988), even though it is not known up to which height this approach is feasible in the MABL. Therefore, additional measurements at coastal and offshore sites complementing those from Vindeby or FINO1 are required to assess the validity of stability-dependant turbulence models for the design of tall offshore wind turbines.

## 7  Conclusions

This study explores the turbulence spectral characteristics from wind records of a year duration on an offshore mast called South

Mast West (SMW) near the first offshore wind farm Vindeby. This study aims to identify similarities between the turbulence characteristics estimated on the FINO1 platform (North Sea) and those at Vindeby. Such an investigation is crucial to establish appropriate turbulence models relevant for the design of offshore wind turbines (OWTs). The dataset analysed was acquired by 3D sonic anemometers at $6\,\mathrm{m}$, $18\,\mathrm{m}$ and $45\,\mathrm{m}$ above the mean sea level (amsl), which complements the dataset collected between $40\,\mathrm{m}$ and $80\,\mathrm{m}$ amsl on the FINO1 platform (Cheynet et al., 2018).

Although the non-dimensional mean wind speed profile seems to follow Monin-Obukhov similarity theory (MOST) between $6\,\mathrm{m}$ and $45\,\mathrm{m}$ amsl at SMW, the turbulence characteristics at $6\,\mathrm{m}$ amsl showed deviations from MOST, especially under neutral and stable conditions. The friction velocity measured at this height is substantially higher than at $18\,\mathrm{m}$ amsl and $45\,\mathrm{m}$ amsl. In this regard, the measurements at $6\,\mathrm{m}$ could be considered to be located in the wave sublayer, although a direct correlation between the sea surface elevation and the turbulent fluctuations was observed for significant wave heights above $0.9\,\mathrm{m}$ only.

This correlation was quantified in terms of co-coherence between the vertical turbulence component and the velocity of the sea surface. The dominant contributor for the larger friction velocity at $6\,\mathrm{m}$ amsl may be the heterogeneous surface roughness that increases in the proximity of SMW.

The velocity records at $18\,\mathrm{m}$ amsl follow fairly well the surface-layer scaling and their spectral characteristics are consistent with those from the measurement at FINO1 platform at $40\,\mathrm{m}$ amsl (Cheynet et al., 2018). Because the measurements at $6\,\mathrm{m}$ and

$18\,\mathrm{m}$ amsl are located in the lower part of the surface layer, a wide spectral peak at near-neutral stratification is observed, which reflects the distortion of the eddies as they scrape along the surface. The turbulence spectra at $45\,\mathrm{m}$ amsl agree reasonably well with those from the FINO1 platform under convective conditions. For neutral and stable conditions ($-0.1 \le \zeta \le 1$), both spectra at $45\,\mathrm{m}$ on SMW and $41.5\,\mathrm{m}$ at FINO1 are especially similar, except in the high-frequency range, where measurement noise is prevailing at $45\,\mathrm{m}$ on SMW.

The co-coherence estimates of the along-wind component for neutral atmospheres are exquisitely well-described by the same 3-parameter exponential decay function as used at FINO1 (Cheynet, 2019). However, this is not the case for the cross-wind and the vertical wind components due to the closer distance to the sea surface.

The comparison between the turbulence data at Vindeby and FINO1 is valuable to further develop spectral turbulence models that are suitable for modern OWT designs. Nevertheless, the wind load designs require the knowledge of turbulence



characteristics at heights up to $200\,\mathrm{m}$, which was not achieved at SMW nor FINO1. It is therefore, necessary for future atmospheric measurements to cover this height, where surface-layer scaling may no longer applicable.

*Code and data availability.* The codes can be made available upon request. The raw data was provided by an external party and therefore
cannot be made available.

*Author contributions.* RMP and EC provided the data curation, formal analysis, methodology, software, and visualization. RMP and EC prepared the original draft. JBJ and CO provided the supervision, review, and editing of the manuscript. All authors contributed to the
conceptualization and finalization of the paper.

*Competing interests.* The authors declare that there are no conflict of interests present.

*Acknowledgements.* The authors would like to express their gratitude to Dr. Kurt Hansen for providing the Vindeby dataset. Professor Joachim Reuder and Dr. Stephan Kral are acknowledged for the fruitful discussion on the wave and wind conditions near SMW.



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
