# Peer review of "Turbulence in a coastal environment: the case of Vindeby"

_Wind Energy Science, 2021_

## Community Comment (CC1)

**Comments on:**

**"Turbulence in a coastal environment: the case of Vindeby" :Rieska Mawarni Putri, Etienne Cheynet, Charlotte Obhrai and Jasna Bogunovic Jakobsen**

By Jørgen Højstrup, jorgen@hojstrup.eu

The very thorough review by the anonymous reviewer covered most of my comments also. I just want to add a few items:

- Data from additional instrumentation available
- Sonic anemometer flow distortion
- Tower flow distortion
- Spikes in data

**Background**

I was responsible for running the Vindeby measurements and I had a lengthy email correspondence with the primary author on the measurements and the corresponding databases of analysed data and timeseries in the spring of 2020.

**Additional instrumentation**

During two prolonged periods in 1994 additional instrumentation was added to the experiment called the RASEX measurements [1]. Instrumentation included wave wires for more accurate (and better special resolution) wave measurements and three additional sonic anemometers of which two were of a type with considerably less flow distortion.

**Sonic anemometer flow distortion**

In addition to the Solent omni-directional anemometers during the basic measurements, three more sonics were added, at 32m (omnidirectional Solent), 3m and 10m (asymmetric Solent with less flow distortion). The omnidirectional Solent shows considerable flow distortion, here illustrated with the ratios of measured u* at 18m (omnidirectional) to measured u* at 10m (asymmetric sonic), fig. taken from a presentation by me at Oregon State University, 7 May, 1998:

[Figure]

**Tower flow distortion influence on phi_m**

When calculating phi_m with measurements from a tower like the one used in Vindeby, you need to take into account the variation with height of the flow distortion caused by the tower (fig. 6 in [2]). There were anemometers on both sides of the mast which enabled modelling of the flow distortion and its influence on the wind profiles [2].

Furterhermore it was shown that phi_m varies with sea fetch [2], which was also not taken into account in the WES paper:

[Figure]

***Figure 6*** *Nondimensional wind profiles as a function of stability. The full line is long fetch data, whereas the dashed line is short fetch data. The model is shown as a dotted line.*

**Spikes in data**

On page 9 the authors refer to a fairly crude method for removing spikes. Checking for spikes using a much better method [3] was part of the QC routine and the data analysis – and of course filtering out data with strong precipitation left data with very small amount of spiking (precipitation sensor on mast LM).

[1] Højstrup, J., J. Edson, J. Hare, M. S. Courtney, and P. Sanderhoff, 1997: The RASEX 1994 experiments. Risø-R-788, Risø National Laboratory, Roskilde, Denmark.

[2] J. Højstrup: Vertical extrapolation of offshore wind profiles. In proceedings from EWEA conference in Nice, France, 1999.

[3] J. Højstrup: A statistical data screening procedure. Meas. Sci. Technol. 4 (1993) pp. 153-157

---

## Author Comment (AC2)

**Answer to reviewers**

Dear Sir/Madam,

We thank the reviewers for the feedback to better improve our manuscript. Below is our answer to the reviewer's comments.

**Reviewer 1**

**Reviewer's summary:**

The manuscript attempts to analyze sonic measurements at one of the masts of the Vindeby wind farm to describe the turbulence characteristics in the offshore marine boundary layer. As it is shown, I have some major comments and a number of minor comments. I think the manuscript has some potential but right now it reads more as an overly descriptive technical report than a journal publication.

**Overall response:**

The original draft of the manuscript, especially the abstract may mislead the reader and has now been reformulated in the revised version. To make the manuscript more concise, we have created appendices to accommodate some of the supporting information that were detailed in the main body of the original submission. The definition of the Monin-Obukhov similarity theory (MOST) and the surface-layer scaling have been re-evaluated carefully in the revised manuscript. Concerning the flow distortion issue, it is addressed in detail in the reply to the community comment by Dr. Højstrup.

**Major comments**
* * *
***Q 1.1*** *The manuscript is too long and overly descriptive. I understand that the authors think that the many different aspects they are trying to analyze deserve to be documented/published. However, this effort makes the text to be tedious and too extensive. Also there is the tendency to explain concepts/theories that do not need explanation. Most importantly, the overly descriptive and long aspect makes the study too unfocused and so the manuscript reads more as a technical report of different analyses, which were performed with these measurements during the course of a project. I suggest that the authors concentrate in a particular aspect (I will suggest later which one(s)) and develop the manuscript towards answering the questions that such aspect(s) rises.*

**Reply**: In the attempt to make the manuscript easier to read, we have moved some of the results in the appendix. The objective of the paper is to identify similarities between the spectral characteristics at FINO1 and Vindeby. The paper focuses on three aspects:

- A data quality assessment of the sonic data

- The one-point spectral characteristics with a comparison from the predictions from FINO1

- The two-point spectral characteristics through the co-coherence with another comparison with the predictions from FINO1

In this regard, the manuscript is quite specific and focused on the flow characteristics required for wind loading on offshore wind turbines. We are aware that we are quite detailed in the data processing and data quality analysis but we consider a stringent and well-documented data analysis of a study dealing with full-scale measurements.

Finally, we explain concepts and theories, even already known, these were found necessary to make the manuscript self-explanatory. The manuscript targets an audience with an engineering background (since it is about wind energy) so we feel it is necessary to be as pedagogical as possible.
* * *
***Q 1.2*** *In general, and particularly in the abstract, the authors claim that spectra follow/not follow MOST.*

*MOST does not really predict the behavior of the velocity spectra. MOST basically says that within the surface layer, gradients (wind and temperature) when properly normalized (e.g. by u\*) are function of the dimensionless stability. Yes, one can also prove that similarly to MOST, proper scaling can be applied to the spectra but this does not mean that MOST itself suggested such scaling for the spectra. I guess you can call it "surface-layer scaling" or "surface-layer similarity".*

**Reply**: Our reference was to Kaimal et al. (1972) and his contribution to MOST, i.e. we did not imply that Monin and Obukhov suggested the scaling of the velocity spectra in the surface layer. To keep it consistent with the wording used by Kaimal and Finnigan (1994), in the revised manuscript, when we talk about the spectral characteristics, we now refer to "surface-layer scaling" instead of "MOST".
* * *
***Q 1.3*** *In the abstract the term "turbulence characteristics" is used. You should specify what do you mean by this. Is it about length scales? Spectral peaks? Turbulence anisotropy or dissipation? What are the characteristics you are referring to?*

**Reply**: We are referring to the one-point auto spectral densities and the real part of the root-coherence. These are now clarified in the abstract.
* * *
***Q 1.4*** *In the last part of the abstract and I think later in the conclusions, the authors mention that their findings are relevant for load estimations of offshore wind farms. However, as the authors acknowledge, the levels they study do not cover those in which current offshore turbines operate. So what did we learn for offshore wind energy? In combination with my first point, I think that the authors can concentrate on understanding aspects we have not explored much in wind energy although they might not have an impact on the loads of turbines (I am fine whether this is important or not to loads). For example, I was particularly happy to see that they were looking at the influence of waves on the turbulence measures. However, I was disappointed because the authors do not seem to make an effort on continuing analysing this influence. For me, it seems to be the most interesting aspect that the paper could explore and I would recommend that a future revision of manuscript focuses on this*

**Reply**: Coherence is a key parameter for wind loading on structures. In this study, the similarities we identify with the one estimated at FINO1 support our statement that our findings are highly valuable for wind turbine design.

Our findings regarding the one-point velocity spectra for wind turbine design are also crucial for any researcher working on (offshore) wind loading because such a description is rarely available in the scientific literature. Furthermore, since we use surface-layer scaling, the description of the power-spectral densities is adapted for the entire surface layer. Also, we highlight multiple similarities between the velocity spectra at Vindeby and FINO1, which is one of the key aspects needed for offshore wind turbine design.

It should be pointed out that the study of wind-wave interaction is limited by the available instruments near SMW. Also, the paper is about the characterization of atmospheric turbulence for wind turbine design. Elaborating excessively on the air-sea interaction would mean that the paper goes off-topic and that it would not be suitable any longer for Wind Energy Science.
* * *
***Q 1.5*** *You are using Gill 3-axis sonics. These are known to be affected by flow distortion. Do you apply any flow distortion correction to these measurements? If not, why not? I think you should elaborate*

*more on this as you also point out (see line 272) that w in particular could be highly affected by probe-induced flow distortion. So if there is flow distortion (I think there is) why will this affect more the 6 m than the 18 or 45 m measurements? By how much you will reduce or increase your fluxes using corrections for flow distortion (in relation to your quest on finding out the differences between $u_*$ at the different levels)? Until this is not clarified, then I would omit Figs. 4 and 5 (and so help a little bit with shortening the paper as part of my comment 1). You kind of "deny" the flow distortion issue by saying that your Sw/Su ratios (1.2) are close to the ones of Fino. This is however not an argument as at Fino there might be other things happening and the same ratio can be achieved by the combination of two opposite issues, for example (two or more wrongs can make the result to look good). As you also mention (lines 297-298) the spectral ratios are more easily reached by Sv/Su than Sw/Su, which is a sign of flow distortion!*

**Reply**: We have added a new section in the appendix to address the flow distortion by the sonic anemometers. More details about this topic are given in our reply to Dr. Højstrup, who wrote a community comment (CC) available at https://doi.org/10.5194/wes-2021-75-CC1. The statement of the reviewer that Gill 3-axis sonic anemometers are known to be affected by flow distortion is partly true only. It has been documented for more than one decade that all sonic anemometers are affected by flow distortion (Peña et al., 2019; Horst et al., 2015). If the reviewer refers to the Gill "w" bug (Instruments, 2016), this only applies to a number of Gill anemometers produced between 2006 and 2015. Flow distortion is partly corrected during wind tunnel tests at the time of the instrument's calibration. A posteriori correction is not straightforward unless the sonic anemometer is tested again in a wind tunnel.
* * *
**Q 1.6** *Line 278: friction velocity averaged between two heights in Fig. 5? I guess you mean Fig. 6?*

*But anyway, you should not do that. How are you computing du/dz? Simple wind speed differences between heights? You seem to have the opportunity to use the cup anemometers that are just above and below the sonic to do this (and avoid please the friction velocity averaging). So these dimensionless wind shears need to be recomputed. If the cup data is not there then you should use at least three speed levels to do a better fitting, e.g. using a wind speed polynomial but still using the local friction velocity. And yes, measurements below 10 m might be outside the surface layer but inside the viscous or wave layer (in the offshore case). So it is actually ok to find that $\phi_m$ at this heights are not following MOST.*

**Reply**: The results presented in the original draft showed that $\phi_m$ follows fairly well MOST but deviations from MOST were found for $\phi_w$. An improved agreement of $\phi_w$ with MOST was obtained by using local surface-layer scaling, which was not the case in the original draft. In the revised manuscript, the discussion about $\phi_w$ was removed, as suggested by the reviewer for the following reasons: (1) there exist some uncertainties due to transducer-induced flow distortion, and (2) the study of $\phi_w$ is not among the main objectives of the paper.

Regarding the study of $\phi_m$, we thank the reviewer for the different suggestions. We have tested different approaches to compute $\phi_m$ using a polynomial fit with an order of 1 or 2 to the wind speed gradient and the friction velocity to obtain local values. In the present case, we found that the approach presented in the original draft was more appropriate. The polynomial fit led to some spurious results.

Regarding the choice of friction velocity as an average or a local measurement, the choice is not as straightforward as implied by the reviewer. The friction velocity is responsible for the largest uncertainty in the calculation of $\phi_m$ since it is derived from the covariance of two weakly correlated time series. Therefore, the use of spatial averaging reduces the uncertainties. If the friction velocity changes significantly with the height, it may reflect the presence of an internal boundary layer. In this case, $\phi_m$ is not defined.

In this study, we decided to focus on the sonic anemometer data as the cup anemometer records have been analysed in earlier studies. In our investigation of the wind-wave interactions, we have demonstrated that the number of samples at 6 m located in the wave sublayer is not significant. Since

the manuscript is about the comparison of the spectral characteristics at Vindeby with the predictions from FINO1, we do not think it is necessary to further elaborate on the different approaches to compute $\phi_m$.
* * *
***Q 1.7*** *I am not sure if the amount of records you are using to derive the spectra (Figs 12-14) are the same that you use to present the other stability-related results, but when looking at these figures I can see that your records in the most unstable and stable cases are too few and too noisy particularly in the very stable plot. So I recommend you do not use those stability bins and I recommend you combine the next two stables ones in one and the next two unstable ones in one to increase the significance of the results and reduce the noise.*

**Reply**: The number of samples for $\zeta > 1$ is 18, which is good enough to obtain meaningful averaged spectral characteristics. For comparison, Kaimal et al. (1972) had only five hours of data for stable conditions and could still draw useful conclusions. The number of samples is not the only parameter that plays a role in the estimation of the averaged power spectral density. Knowledge of the method to compute the power spectral densities is also needed. In the present case, we used Welch's modified periodogram with three segments, which "artificially" increases the number of samples. We do not recommend merging the stability bins, given that the noise level at 18 m and 45 m increases with the stability. Furthermore, the turbulence can change significantly for $0.5 < \zeta < 2$. For this reason, we decided to keep the 9 panels and simply warn the reader that with fewer samples, the uncertainties are larger.

**Minor comments**
* * *
***Q 1.1*** *Line 2: The second line should read "Sonic anemometer measurements at 6, 18 and 45 m . . .", so that we already know you are using sonic observations. Also for this and all instances, compact the listing: so instead of saying "6 m, 18 m and 45 m" replace by "6, 18 and 45 m"*

**Reply**: Replaced, as suggested by the reviewer.
* * *
***Q 1.2*** *Line 6: replace "empirical spectra established on" by "that from"*

**Reply**: Replaced, as suggested by the reviewer.
* * *
***Q 1.3*** *Line 9: Replace "with those at" by "that at"*

**Reply**: Replaced, as suggested by the reviewer.
* * *
***Q 1.4*** *Line 37: Replace "are justified for" by "relate to"*

**Reply**: Replaced, as suggested by the reviewer.
* * *
***Q 1.5*** *The sentences between lines 39 and 43 need to be rewritten. First I thing you are mainly talking about the Mann model and second I am not sure of what model you refer to when citing Kelly (2018) (from what I can see there is no model there other than the Mann model)*

**Reply**: We agree with the reviewer. However, we prefer the use of "uniform shear model" rather than "Mann model" because there are two different models proposed in Mann (1994): the uniform shear (US) and the uniform shear with blockage by the surface (US+B). These sentences have been reformulated as: "In this regard, the present study addresses similar challenges as discussed by Kelly (2018) but

focuses on some specific aspects not covered by the spectral tensor of homogeneous turbulence (Mann, 1994). Firstly, the low-frequency fluctuations are generally underestimated by the uniform-shear model, especially under convective conditions (De Maré and Mann, 2014; Chougule et al., 2018). Secondly, the vertical coherence of turbulence is not always described accurately by the spectral tensor (Mann, 1994; Cheynet, 2019)."
* * *
**Q 1.6** *Lines 48 and 50 and maybe other instances: be consistent so it is either heights, levels or altitudes (first is preferable)*

**Reply**: The term 'height(s)' is now used in the revised manuscript whenever possible for consistency.
* * *
**Q 1.7** *Line 45: "semi-empirical models from FINO1": this is the first time we hear somebody came up with such models from FINO1, so you need to provide some context, a reference, and probably also say models of what exactly*

**Reply**: We have reformulated the sentence as: "Then, the one-point velocity spectra and co-coherence estimates from Vindeby are compared with predictions from semi-empirical models established on the FINO1 platform (Cheynet et al., 2018) to assess the similarities of the spectral characteristics between the two sites."

This sentence implies directly that the semi-empirical models we are referring to are the one-point velocity spectra and the coherence. The reference is deemed adequate so that we do not need to elaborate further on the model. Otherwise, the manuscript may become too large.
* * *
**Q 1.8** *Figure 1: Denmark in the left and particularly Lolland in the right look quite flooded (blue areas where green should be). I guess this is because your Digital Elevation Model (DEM) shows 0 m for areas that are not water areas*

**Reply**: Yes this is a correct guess. We have updated the figure with the corrected land cover.
* * *
**Q 1.9** *Lines 65 and 84 delete "of" after "comprised"*

**Reply**: Deleted, as suggested by the reviewer.
* * *
**Q 1.10** *Section 2: I do not think you mention what kind of cups and vanes you have and the heights where they measure*

**Reply**: The measurements from the cup anemometers were not used, therefore we did not provide the cup anemometers' detail in the manuscript. To prevent confusion, we added a statement in the manuscript to clarify that the measurements from the cup anemometers were not used, which reads as "There were seven cup anemometers mounted on SMW as shown in fig. 3. However, their measurements were not used here". The heights of the vanes are already provided in the following line "Two Risø P2021 resolver wind vanes with wind direction transmitters P2058 were located on the northern booms at 43 m and 20 m amsl using a sampling frequency of 5 Hz".
* * *
**Q 1.11** *Line 82: delete "the wind"*

**Reply**: Deleted, as suggested by the reviewer.
* * *
**Q 1.12** *Line 94: add "as" after "denoted"*

**Reply**: Added, as suggested by the reviewer.
* * *
**Q 1.13** *Line 96: Replace "To study turbulence for wind turbine design" by "Here,"-> this is an example of lengthy sentences that can be shortened without deteriorating and makes the paper shorter (there are many like this so please make an effort to be more concrete and short)*

**Reply**: Replaced, as suggested by the reviewer. The other changes are marked with magenta-coloured fonts.
* * *
**Q 1.14** *Line 98: "modeling the v-component" I guess you mean modeling the v-spectrum or v-variance as v=0 in most cases as we align u with the mean wind (you do that actually)*

**Reply**: What we meant by modelling the *v*-component is the time histories of the lateral wind velocity component. This implies the modelling of both its one-point power spectral densities and its root-coherence. This has been clarified in the following line "Although the *u*-component drives the wind turbine's rotor fatigue loads, proper modelling of the *v*-component in terms of power spectral density (PSD) and root-coherence may be necessary for skewed flow conditions, which can occur because of a large wind direction shear (Sanchez Gomez and Lundquist, 2020) or wind turbine yaw error (Robertson et al., 2019)".
* * *
**Q 1.15** *Line 110: add "vertical" before "flux"*

**Reply**: Added, as suggested by the reviewer.
* * *
**Q 1.16** *Line 113: why is $\theta_v$ not reliably measured by a sonic?*

**Reply**: The mean sonic temperature was known to have a measurement bias. This was documented to us through private communication with Dr. Kurt Hansen. However, the fluctuating sonic temperature is unaffected by such a bias since, per definition, the fluctuating component is detrended. The mean virtual potential temperature was thus obtained using other sensors, as described in the manuscript.
* * *
**Q 1.17** *Line 121: $\phi_w$ is not commonly used to assess MOST. Perhaps $\phi_m$ and $\phi_{temp}$*

**Reply**: As emphasised by e.g. De Franceschi et al. (2009), $\phi_w$ has become widely used in the past 30 years to study the applicability of MOST. The study of $\phi_w$ has a considerable advantage over methods based on a gradient or bulk parametrisation as it allows using point-measurements with ultrasonic anemometers. The use of 3D sonic anemometer in the 1990s has made highly relevant the study of the ratio $\sigma_w/u_*$ as a function of $z/L$ to assess the applicability of MOST.
* * *
**Q 1.18** *Line 135: the spectrum is not a quantity. Anyway, the whole paragraph between lines 135 and 138 is not needed*

**Reply**: This line is formulated as "An appropriate modelling of the one-point velocity spectrum is required to compute reliably the dynamic wind-induced response and the power production of wind turbines (Sheinman and Rosen, 1992; Hansen and Butterfield, 1993)" in the revised manuscript. The lines 135-138 are not omitted since they provide an important point that is not addressed in the IEC 61400-1. Turbulence intensity (TI) is considered as an important input for turbulence modelling according to IEC 61400-1. Nonetheless, as pointed out by Wendell et al. (1991), turbulence intensity may not always be a reliable characterization of turbulence because TI does not carry information concerning the distribution of eddies in the frequency domain.
* * *
**Q 1.19** *8: remove the 2/3 as exponent of $\phi_\varepsilon$*

**Reply**: Removed, as suggested.
* * *
**Q 1.20** *12-14: Between the description of these equations you should give some values for c1 and c2 so that c in Eq. 12 is negative otherwise you need to add a minus in the argument of the exponent*

**Reply**: A minus sign is added in Eq. 12.
* * *
**Q 1.21** *15 is the cross a dot product?*

**Reply**: It is not a cross but an ordinary multiplication symbol. The 'cross' symbol has been removed from Eq. 15 to avoid confusion.
* * *
**Q 1.22** *Line 197: so is data plural or singular?*

**Reply**: According to Copernicus Publication, the word "data" is considered as a countable noun (e.g. data are, data were, data include). In the revised manuscript, we evaluate the word "data" as countable noun.
* * *
**Q 1.23** *Line 200: reliable estimation of Obukhov length means turbulence flux estimations. Why not completely taking out the 45 m sonic anemometer measurements, at least for the spectra analysis? I mean you continuously mention that this sonic is highly affected by noise. For the coherence it could be fine to use as the noise reduces by the cross-spectrum computation.*

**Reply**: We also mention that the measurement quality is satisfying for wind speed above 8-10 m s$^{-1}$ at this height. Therefore, it would be inaccurate to state that the measurement noise prevents any analysis. The reviewer's comment regarding the noise reduction by the normalisation of the cross-spectrum for the study of the coherence is correct. The co-coherence is not substantially affected by the noise because the noise is more important at high frequencies, where the co-coherence is typically zero for the separation distances considered.
* * *
**Q 1.24** *Line 212: how do you know the planar fit gives better estimates of covariances compared to double rotation? I mean compared to what? In my understanding, it is completely the opposite*

**Reply**: We did not conclude that the planar fit method gives better estimates of covariances, it was simply stated in the work by Wilczak et al. (2001). To avoid confusion, we added a reference to the work by Wilczak et al. (2001) in this sentence. The sentence is now read as"It should be noted that this finding is likely specific to the Vindeby data-set as the planar fit method usually provides better estimates of the turbulent fluxes (Wilczak et al., 2001)".
* * *
**Q 1.25** *Line 234: to compute that mean wind speed you need also a friction velocity value at least (which you do not mentioned) or need to do perform another computation/assumption (such as a geostrophic drag law)*

**Reply**: We did compute the mean wind speed using the following relation:

$$u_{z2} = \frac{\ln(\frac{z_2}{z_0})}{\ln(\frac{z_1}{z_0})} u_{z1} \tag{1}$$

where $z_2$ is taken as 90 m (hub height) and $z_2$ is taken as 18 m. The value of $z_0$ was not calculated but taken from the scientific literature (it is a widely used value roughness for calm sea). To avoid further

misunderstanding, we have added some references to justify the use of $z_0 = 0.0002$ m in the manuscript, for example WMO (1983).
* * *
**Q 1.26** *Line 241: delete "which testified"*

**Reply**: Deleted, as suggested by the reviewer.
* * *
**Q 1.27** *Line 265: flat or uniform "terrains" no plural... not the only instance with this issue similar happens with "noises"... no need for plural (line 267 and maybe other instances)*

**Reply**: Changed, as suggested by the reviewer.
* * *
**Q 1.28** *4: friction velocity "computations" or "calculations" not "estimations". Also the dimensionless stability in the legend appears with units of m*

**Reply**: The friction velocities were computed, however, these numbers are an 'estimation' since we are working with times series that are described as ergodic, stationary random processes. We applied temporal averaging operators. Therefore, all statistical quantities computed are, indeed, biased or unbiased estimates of the "true" quantities. The unit m for the dimensionless stability is removed from fig. 4.
* * *
**Q 1.29** *5: For 18 and 6 m, the error bars are quite small the more stable or unstable (the most unstable is nearly zero error for the 6 m). So the uncertainty should be presented with other metric (standard error or deviation). I would definitively skip this graph as the w component might be too affected by flow distortion*

**Reply**: We removed this figure in the revised manuscript as suggested by the reviewer.
* * *
**Q 1.30** *7: units in m/s or in m s$^{-1}$*

**Reply**: For consistency, the unit is re-written to m s$^{-1}$.
* * *
**Q 1.31** *Line 285: the sentence does not makes sense as the 18 m value is a local value*

**Reply**: Initially, the term 'local value' used in this sentence refers to the friction velocity values at each corresponding height. Nonetheless, this sentence has been removed in the revised manuscript because it is related to fig. 5 (variation of $\sigma_w/u_*$) which has been taken out in the revised manuscript, as suggested by the reviewer (see Q 1.29).
* * *
**Q 1.32** *5.3 is not needed and can be removed without endangering the study (see major comment 1)*

**Reply**: We move Section 5.3 (Estimation of the friction velocity) to appendix for shortness and simplicity of the manuscript.
* * *
**Q 1.33** *Line 340: did you introduce the quad-coherence before? I mean you do introduce the Co-coherence*

**Reply**: We did not introduce the quad-coherence, therefore a brief wording is added to introduce the quad-coherence in this line, "The interactions between wind turbulence and the sea surface were explored in terms of the co-coherence and the quad-coherence (the imaginary part of the root-coherence) between the vertical velocity component $w$ and the velocity of the wave surface $\dot{\eta} = d\eta/dt$".
* * *
**Q 1.34** *Line 378: didn't you already introduce the reduced frequency?*

**Reply**: Yes, the reviewer is correct. The reduced frequency $f_r$ is already introduced in the first paragraph of Subsection 3.2, therefore the definition of $f_r$ in this line is now removed in the revised manuscript.
* * *
**Q 1.35** *Line 379: I guess you need to delete the 2/3*

**Reply**: The spectra were normalised by $\phi_\varepsilon^{2/3}$, therefore the "2/3" is not removed, but in Eq. 8 which it was referred to is now deleted, kindly refer to minor comment Q 1.19.
* * *
**Q 1.36** *Line 382: "empirical model established at Fino1... (Cheynet et al., 2018)". So you have not introduced this. If you refer to Eq. (15) you actually attributed this to Cheynet 2019. By the way this also points me to the references: you have way too many references to your own work (Cheynet) and I am sure many others have done similar studies. I am also sure that you do not need to cite all of your studies but a couple of them*

**Reply**: In our manuscript, the number of self-citations is around 5%, which is well below the median value of self-citation documented in the literature, that is between 10% and 13% (Ioannidis et al., 2019; Szomszor et al., 2020). We refer often to the same references because we are conducting a comparison with predictions from another dataset. The reviewer is welcomed to suggest other relevant references to the topic that we may have missed.
* * *
**Q 1.37** *Line 384: "the behavior of surface-layer spectra" does not the spectra of velocities above surface layer also behave like this in the asymptotic limit?*

**Reply**: Not necessarily, because the red curve is obtained for surface layer scaling only. Another scaling is needed above the surface layer. Kindly refer to Section 2.6 from Kaimal and Finnigan (1994).
* * *
**Q 1.38** *Line 387: "which is another... properly" as mentioned earlier two or more wrong things can make the result to be ok so no this is not an indication that the estimation is properly done*

**Reply**: We agree with the reviewer. We have removed this statement from the line in the revised manuscript.
* * *
**Q 1.39** *Sentences between lines 387 and 391 can be removed without detriment*

**Reply**: We agree that the sentences in line 388 to 391 may be repetitive, therefore these lines are now removed.
* * *
**Q 1.40** *In page 19 there are many entries with reference to goodness of the spectra with respect of MOST so this needs to be rewritten (major comment 2)*

**Reply**: We have replaced the word "MOST" with "surface-layer scaling", as detailed in our reply to major comment Q 1.2.
* * *
**Q 1.41** *Line 414: the reasoning of the flatness of the spectral peak is not the difficulty in estimate the integral length scale... on the contrary it is difficult to estimate the length scale due to the flatness of the spectra*

**Reply**: We agree with the reviewer. This sentence was poorly written. We have reformulated it as "This leads to a flat spectral peak. As a result, the integral length scale would be estimated with large uncertainties".
* * *
***Q 1.42*** *Line 418: Well this is nice that you state that these deviations are due to the contribution of waves, but how do you know this? Following my major comment 4, this could be something to concentrate efforts in the study; I mean demonstrating that these deviations are caused by the waves*

**Reply**: In the manuscript, we demonstrate that the number of samples showing a clear correlation between turbulence and the waves is too small to be significant. We cannot demonstrate that the roughness upstream of the mast is due to a heterogeneous sea state. Therefore, we concluded that flow distortion may explain (at least partly) the variability observed, which is a safer assumption.
* * *
***Q 1.43*** *Fig 12 and similar: delete the "for references. . . in the data" of the caption.*

**Reply**: We agree with the reviewer. These are now removed from the caption for fig.10, fig.11, and fig.12.
* * *
***Q 1.44*** *Line 423: is the -2> not a -2<?*

**Reply**: It is corrected to $-2 \leq$.
* * *
***Q 1.45*** *Line 433: delete "since we aim. . . Vindeby"*

**Reply**: Deleted, as suggested by the reviewer.
* * *
***Q 1.46*** *Line 435: did you omit the value of C3v?*

**Reply**: No, it was not omitted. The values are [0, 23, 0.09], however it seems that we missed a coma after "23". To be consistent, we added a coma after 23, so it now becomes [0, 23.0, 0.09].
* * *
***Q 1.47*** *Line 439: so why is the coherence of v negative? It seems to be also the case in Fino1. So, why don't you use the Mann coherence here? I think it could provide you with negative coherences*

**Reply**: The co-coherence of the *v*-component is negative because the vertical shear introduces a time lag between measurements at different heights. The lateral component is more affected than the along-wind component (Bowen et al., 1983). Chougule et al. (2012) offers a possible interpretation for the larger phase angle for the cross-wind component compared to the along-wind component. The negative part is more visible if measurements are located close to the ground, where the mean shear is larger. For this reason, the negative values of the co-coherence were more visible in the Vindeby database than in the FINO1 database. At FINO1, the negative part was small enough to be neglected. The co-coherence estimated with the uniform shear model (Mann, 1994) is, indeed, able to model the negative part of the co-coherence. However, it is unclear whether this model can reliably model the negative part. On the other hand, the uniform shear model is known to overestimates the co-coherence at vertical separations and low frequencies (Mann, 1994). This drawback is of greater importance for wind turbine wind-induced load predictions.
* * *
***Q 1.48*** *Caption Fig. 15: change "empirical values computed" by "predictions"*

**Reply**: Changed, as suggested by the reviewer.
* * *
***Q 1.49*** *Line 447: "lateral co-coherence is also required". . . . For what?*

**Reply**: The sentence is now reformulated as "Finally, additional data collection is needed to study the co-coherence at lateral separations, which is required for wind turbine design since it was not available at FINO1 nor SMW".

A wind turbine cannot be approximated as a line-like structure like a tower or a bridge. Therefore, information on the coherence for both lateral and vertical separation distances is needed. Whereas the vertical coherence can be studied using anemometers mounted on a single mast, the study of the lateral coherence requires either multiple masts or excessively long booms, which may not be feasible offshore for financial or technical reasons.
* * *
**Q 1.50** *Lines 448-450 can be deleted without detriment*

**Reply**: Based on the response for minor comment Q 1.49, these lines are not deleted but reformulated instead: "Further studies are, however, needed to better quantify this possible overestimation in terms of dynamic wind loading on the wind turbine's rotor and tower, as well as on the floater's motions in the case of a floating wind turbine. Finally, additional data collection is needed to study the co-coherence at lateral separations, which is required for wind turbine design since it was not available at FINO1 nor SMW."
* * *
**Q 1.51** *Lines 454-455: do people use vertical coherence models for aeroeslatic turbine simulations when not using the Mann model?*

**Reply**: Yes, people use the vertical (and lateral) coherences when simulating a spatially correlated turbulent wind field when not using the Mann's model. For example, the traditional approach for turbulence generation method by Veers (1988) requires knowledge of the coherence of turbulence.

We have a lot of respect for the uniform shear (US) model, especially regarding its remarkable ability to describe the second-order structure of homogeneous turbulence with a limited set of parameters. However, as mentioned in the introduction, there are several limitations of the US model for aeroelastic loading calculation. One of them is the limited ability of the US model to describe realistically the vertical coherence. On the other hand, the US model is known to perform well when it comes to describing the lateral coherence. In this regard, the US model could still be used to complement the semi-empirical models we have mentioned in the manuscript. Another limit is that the three parameters of the US models do not change in space, which is known not to be the case in reality.
* * *
**Q 1.52** *Line 459: replace "The first one is related to the fact that the" by "The"*

**Reply**: Replaced, as suggested by the reviewer.
* * *
**Q 1.53** *Line 460: why nonstationary time series are not reliable?*

**Reply**: We are estimating turbulence characteristics from time histories using operators such as mean, standard deviation, or power spectral densities. A fundamental condition to use these operators is that the time histories are stationary. Otherwise, the turbulence characteristics are biased. More generally, turbulence is here described as a stationary random process. The description of the non-stationary characteristics of turbulence is beyond the scope of the present study. It can be noted that in standards and codes, the design of offshore wind turbines also relies on the assumption that turbulence is stationary.
* * *
**Q 1.54** *Lines 466-467 Delete "Therefore,... factor"*

**Reply**: Deleted, as suggested by the reviewer.
* * *
**Q 1.55** *Line 469 what does invariant here mean?*

**Reply**: The term 'invariant with height' in this context means does not change with height. The sentence is rephrased to avoid ambiguity, and read as: "Above the surface layer, the velocity spectra may become

independent of the height above the surface, which is coarsely accounted for in IEC 61400-1 (2005) and suggested by preliminary observations from Doppler wind lidar instruments in coastal areas (Cheynet et al., 2021)"
* * *
***Q 1.56*** *Line 471-477: these lines are not needed*

**Reply**: These lines are deleted, as suggested.
* * *
***Q 1.57*** *In the conclusions you again start to introduce acronyms; this is not needed*

**Reply**: A conclusion is not part of the manuscript's body, so acronyms need to be redefined for the sake of clarity. Intuitively, it also makes sense since many readers of a paper only read the abstract, introduction, and conclusion.
* * *
***Q 1.58*** *Line 482: "relevant for the design of offshore wind turbines"... this is not true (major comment 4). Similar issue in line 504*

**Reply**: The latter opinion is refuted by Veers et al. (2019) since wind loading on offshore wind turbines is included in two of the three great challenges in wind energy.
* * *
***Q 1.59*** *Lines 501-502: well the 45 and 18 m are not that close to the surface*

**Reply**: It was not stated in the manuscript that 18 m and 45 m are close to the sea surface. Instead, it is written 'closer to the sea surface' because in this context, we are comparing 18 m and 45 m at SMW with 60 m and 80 m heights at FINO1.

**References**

Bowen, A., Flay, R., and Panofsky, H. (1983). Vertical coherence and phase delay between wind components in strong winds below 20 m. *Boundary-layer meteorology*, 26(4):313–324.

Cheynet, E. (2019). Influence of the measurement height on the vertical coherence of natural wind. In *Conference of the Italian Association for Wind Engineering*, pages 207–221.

Cheynet, E., Jakobsen, J. B., and Reuder, J. (2018). Velocity spectra and coherence estimates in the marine atmospheric boundary layer. *Boundary-layer meteorology*, 169(3):429–460.

Chougule, A., Mann, J., Kelly, M., and Larsen, G. C. (2018). Simplification and validation of a spectral-tensor model for turbulence including atmospheric stability. *Boundary-Layer Meteorology*, 167(3):371–397.

Chougule, A., Mann, J., Kelly, M., Sun, J., Lenschow, D., and Patton, E. (2012). Vertical cross-spectral phases in neutral atmospheric flow. *Journal of Turbulence*, (13):N36.

De Franceschi, M., Zardi, D., Tagliazucca, M., and Tampieri, F. (2009). Analysis of second-order moments in surface layer turbulence in an alpine valley. *Quarterly Journal of the Royal Meteorological Society: A journal of the atmospheric sciences, applied meteorology and physical oceanography*, 135(644):1750–1765.

De Maré, M. and Mann, J. (2014). Validation of the mann spectral tensor for offshore wind conditions at different atmospheric stabilities. In *Journal of Physics: Conference Series*, volume 524, page 012106. IOP Publishing.

Hansen, A. and Butterfield, C. (1993). Aerodynamics of horizontal-axis wind turbines. *Annual Review of Fluid Mechanics*, 25(1):115–149.

Horst, T., Semmer, S., and Maclean, G. (2015). Correction of a non-orthogonal, three-component sonic anemometer for flow distortion by transducer shadowing. *Boundary-Layer Meteorology*, 155(3):371–395.

Instruments, G. (2016). Software bug affecting 'w' wind component of the windmaster family. *Technical key note, Open File Key*.

Ioannidis, J. P., Baas, J., Klavans, R., and Boyack, K. W. (2019). A standardized citation metrics author database annotated for scientific field. *PLoS biology*, 17(8):e3000384.

Kaimal, J. C. and Finnigan, J. J. (1994). *Atmospheric boundary layer flows: their structure and measurement*. Oxford university press.

Kaimal, J. C., Wyngaard, J., Izumi, Y., and Coté, O. (1972). Spectral characteristics of surface-layer turbulence. *Quarterly Journal of the Royal Meteorological Society*, 98(417):563–589.

Kelly, M. (2018). From standard wind measurements to spectral characterization: turbulence length scale and distribution. *Wind Energy Science*, 3(2):533–543.

Mann, J. (1994). The spatial structure of neutral atmospheric surface-layer turbulence. *Journal of fluid mechanics*, 273:141–168.

Peña, A., Dellwik, E., and Mann, J. (2019). A method to assess the accuracy of sonic anemometer measurements. *Atmospheric Measurement Techniques*, 12(1):237–252.

Robertson, A. N., Shaler, K., Sethuraman, L., and Jonkman, J. (2019). Sensitivity analysis of the effect of wind characteristics and turbine properties on wind turbine loads. *Wind Energy Science*, 4(3):479–513.

Sanchez Gomez, M. and Lundquist, J. K. (2020). The effect of wind direction shear on turbine performance in a wind farm in central Iowa. *Wind Energy Science*, 5(1):125–139.

Sheinman, Y. and Rosen, A. (1992). A dynamic model of the influence of turbulence on the power output of a wind turbine. *Journal of Wind Engineering and Industrial Aerodynamics*, 39(1-3):329–341.

Szomszor, M., Pendlebury, D. A., and Adams, J. (2020). How much is too much? the difference between research influence and self-citation excess. *Scientometrics*, 123(2):1119–1147.

Veers, P., Dykes, K., Lantz, E., Barth, S., Bottasso, C. L., Carlson, O., Clifton, A., Green, J., Green, P., Holttinen, H., et al. (2019). Grand challenges in the science of wind energy. *Science*, 366(6464).

Veers, P. S. (1988). Three-dimensional wind simulation. Technical report, Sandia National Labs., Albuquerque, NM (USA).

Wendell, L., Gower, G., Morris, V., and Tomich, S. (1991). Wind turbulence characterization for wind energy development. Technical report, Pacific Northwest Lab., Richland, WA (United States).

Wilczak, J. M., Oncley, S. P., and Stage, S. A. (2001). Sonic anemometer tilt correction algorithms. *Boundary-layer meteorology*, 99(1):127–150.

WMO (1983). *Guide to meteorological instruments and methods of observation*. Secretariat of the World Meteorological Organization.

---

## Author Comment (AC3)

**Community Comment by Dr. Højstrup**

November 30, 2021

We thank Dr. Højstrup for the feedback on our manuscript. Below is our answer to his comments.

**Sonic anemometer flow distortion**
* * *
*Q 1.1* *In addition to the Solent omni-directional anemometers during the basic measurements, three more sonics were added, at 32m (omnidirectional Solent), 3m and 10m (asymmetric Solent with less flow distortion). The omnidirectional Solent shows considerable flow distortion, here illustrated with the ratios of measured u\* at 18m (omnidirectional) to measured u\* at 10m (asymmetric sonic), fig. taken from a presentation by me at Oregon State University, 7 May, 1998:*

[Figure]

Figure 1: Ratio of the friction velocity estimated at 18 m (omnidirectional Solent) over the one at 10 m (asymmetric Solent) by Dr. Højstrup. Unknown time period.

**Reply**: The point raised by Dr. Højstrup is indeed relevant to the present study. We have added an appendix in the manuscript, to discuss the transducer-induced flow distortion. The following content takes the appendix and complements it when necessary. We remind that the present Community Comment is publicly available, which means that our reply is also available to anyone.

The dataset at 3 m was too short to be meaningful so it is not discussed hereinafter. So we will focus mainly on the asymmetric solent at 10 m. The dataset from the sonic anemometer (SA) at 10 m was from May 1994 to September 1994, which was still much shorter than the other instruments.

In the following, one assumes that the sonic anemometer at 10 m does not show significant flow distortion for the sector 220°-330°. The latter sector is the one that was selected in the

original draft. It is possible to partly correct the friction velocity estimate at 6 m, 18 m and 45 m for the flow distortion by the transducer by using a multivariate regression analysis. The objective of the correction is to assess whether the corrected friction velocity changes substantially the results regarding the power spectral densities of the velocity fluctuations.

In the present case, the flow distortion is assumed to be a function of the angle of attack $\alpha(z)$ and wind direction $\theta(z)$ only. For the relatively narrow sector selected, it was found that cubic functions of $\alpha(z)$ and $\theta(z)$ were sufficient to describe this variability. This leads to the following relationship between the friction velocity at 10 m and the height $z$:

$$u_*(z) = (u_*)_{10} \cdot \mathbf{A}\mathbf{X}^\top \tag{1}$$

where

$$\mathbf{A} = \begin{bmatrix} a_1 & a_2 & a_3 & a_4 & a_5 & a_6 \end{bmatrix} \tag{2}$$
$$\mathbf{X} = \begin{bmatrix} \theta(z) & \theta(z)^2 & \theta(z)^3 & \alpha(z) & \alpha(z)^2 & \alpha(z)^3 \end{bmatrix} \tag{3}$$

The coefficients to be determined with the regression analysis are $a_i$, $i = \{1, 2, 3, 4, 5, 6\}$ as shown by eq. (2). In eq. (1), the error is modelled as a non-linear function of the angle of attack and wind direction. In this regard, we do not assume that the friction velocity is constant with the height nor that the flow distortion is similar for the three omnidirectional anemometers.

In fig. 2, we have reproduced some of the results from fig. 1 but for the sector addressed in the present study, i.e. between 220° and 330°. The left (right) panel shows the uncorrected (corrected) ratio of the friction velocity estimates. Including larger sectors has limited usefulness for this comparison. In particular, there exist sectors where the transducer shadowing is much larger for the asymmetric solent at 10 m than the omnidirectional solent at 6 m, which is not clearly highlighted in fig. 1. Therefore, fig. 1 should be interpreted with caution.

In the left panel of fig. 2, the maximal variations of the friction velocity between the sonic anemometer at 10 m and 18 m are ±20%. When all the samples in the sector 220°-330° are averaged, the relative difference at 6 m, 18 m and 45 m with respect to the data at 10 m are 4%, 12% and 11%, respectively. After the multivariate regression, the mean error was close to zero, although it is clearly not zero for a given wind sector (fig. 2). On average, the friction velocity

[Figure]

Figure 2: Ratio of the friction velocity at 18 m (omnidirectional solent anemometer) over the one estimated at 10 m (asymmetric solent anemometer) before (left panel) and after (right panel) correction using a multivariate regression analysis. Velocity data recorded between May 1994 and September 1994 for the sector 220°-330° were used (480 samples of 30 min duration) and $|z/L| < 2$ at 10 m asl.

[Figure]

Figure 3: Power spectral densities of the along-wind component using the uncorrected friction velocity (top panels) and corrected one (bottom panels). The parameter $z/L$ was estimated at 10 m and the data set relied on measurements from May 1994 to September 1994.

estimates at 6 m and 10 m are, therefore, almost identical, given that the random error on the friction velocity is above 10% for a sample duration of 30 min (Kaimal and Finnigan, 1994).

Using data between May 1994 and September 1995, the power spectral densities of the $u$ component with and without corrected friction velocity is displayed in fig. 3. In this figure, the non-dimensional stability parameter is estimated using the anemometer at 10 m. For convective conditions with $\zeta < -0.3$, the uncorrected data shows a more realistic behaviour than the corrected data at low frequencies, where the spectral curves are not expected to collapse onto each other. For near-neutral conditions, the corrected data deviates from the semi-empirical slope in the inertial subrange, marked in red in fig. 3. For stable conditions, the corrected data shows an improvement of the spectral shapes, but the number of samples is relatively low. When the entire dataset (April 1994-July 1995) is used, the velocity spectra normalized with the corrected friction velocity do not show more realistic behaviour than those normalized with the uncorrected friction velocity.

In conclusion, a method to mitigate the influence of the flow distortion on the friction velocity estimate was applied using a multivariate regression analysis. While the flow distortion by the sonic anemometer at 10 m is likely smaller than for the others, the dataset for this sensor was much shorter. The corrected friction velocity did not clearly indicate that the ensemble-averaged normalized spectra were substantially affected by the flow distortion. Flow distortion seems to be mitigated by the fact we averaged samples from an entire sector (220°-330°). For this sector, both an underestimation and overestimation of the friction velocity may be obtained on the omnidirectional sonic anemometers, depending on the wind direction. This could justify the lower-than-expected discrepancies between the uncorrected and corrected averaged spectral flow characteristics.

**Tower flow distortion influence on $\phi_m$**

**Q 1.2** *When calculating $\phi_m$ with measurements from a tower like the one used in Vindeby, you need to take into account the variation with height of the flow distortion caused by the tower (fig. 6 in [2]). There were anemometers on both sides of the mast which enabled modelling of the flow distortion and its influence on the wind profiles [2]. Furthermore it was shown that $\phi_m$ varies with sea fetch [2], which was also not taken into account in the WES paper:*

[Figure]

Figure 4: Nondimensional profiles as a function of the stability.

**Reply**: In the manuscript, we focused on wind direction between 220° and 330° only, such that the fetch was uniform and at least 15 km. Therefore, the sonic anemometers were not affected by tower shadowing. Also, the choice of this sector implies that, for the heights from 6 m and 45 m above the surface, all the sensors are in the internal boundary layer representative of the sea surface. Therefore, the second comment is not applicable in our present study.

Finally, it should be reminded that a fundamental condition to calculate $\phi_m$ is that there is no discontinuity in surface roughness, i.e. the measurement heights are in the same internal boundary layer. Otherwise, $\phi_m$ becomes meaningless. In fig. 4, there is a clear indication that for the fetch of 1.5 km, the measurement heights are located in different internal boundary layers. In this regard, $\phi_m$ does not satisfy MOST. Therefore, the statement from Højstrup, J. (1999) that $\phi_m$ varies with sea fetch should be interpreted with caution since in this particular case, $\phi_m$ is actually not applicable.

**Spikes in data**
* * *
*__Q 1.3__ On page 9 the authors refer to a fairly crude method for removing spikes. Checking for spikes using a much better method [3] was part of the QC routine and the data analysis – and of course, filtering out data with strong precipitation left data with very small amount of spiking (precipitation sensor on mast LM).*

__Reply__: We thank Dr. Højstrup for the suggestion regarding his algorithm (Hojstrup, 1993), but the conclusion that the despiking approach adopted in the present manuscript is too crude is overhasty. We have, therefore, completed the paragraph on page 9, which now reads as

"The time series were sometimes affected by the outliers. In the present case, outliers were identified using a moving median window based on 5 min window length. The same outlier detection algorithm was also used for the sea surface elevation data, but with a moving window of 180 s. The local median values were then used to compute the median absolute deviation (MAD), as recommended by Leys et al. (2013). Data located more than five MAD away from the median were replaced with NaNs. The generalised extreme Studentized deviate test (Rosner, 1983) was also assessed to detect outliers but did not bring significant improvements. When the number of NaNs in the time series was under 5%, they were replaced using a non-linear interpolation scheme based on the inpainting algorithm by D'Errico (2004) with the "spring" method. A more adequate but slower approach using an autoregressive modelling (Akaike, 1969) was also applied but yielded a similar conclusion and therefore was not used. Time series containing more than 5% of NaNs were dismissed. Although other spike detection and interpolation algorithms exist in the literature (e.g. Hojstrup (1993)), the approach adopted here was found to provide an adequate trade-off between computation time and accuracy."

We clarify under our statement regarding two aspects: (1) the spike detection, and (2) data removal and interpolation. We have evaluated multiple spikes detection algorithms, also called outlier detection algorithms. In particular, we have explored the use of the generalized extreme Studentized deviate test (GESD) as well as the moving median window technique. Both techniques performed equally well but the moving median filter was much faster. So it was adopted. The spike detection technique relies on the median absolute deviation (MAD), which is known to be superior to methods relying on the mean and variance of the time series (Leys et al., 2013). In this regard, the spike detection algorithm by Hojstrup (1993) may be criticized for not relying on the MAD.

When the percentage of detected outliers was under 5%, outliers were removed and interpolated values were used instead. We have explored two interpolation approaches. The first one relies on autoregressive modelling (Akaike, 1969) which is similar to the approach by Hojstrup (1993), although the latter paper does not refer to Akaike (1969). Another approach was explored using the inpainting algorithm by D'Errico (2004). The latter was found to provide acceptable performance compared to the autoregressive model while being considerably faster. Velocity data heavily affected by precipitation are associated with the non-Gaussian distribution or time-variable characteristics which are filtered out in the stationary tests and study of the kurtosis and skewness of the velocity data.

In conclusion, our outlier and peak detection algorithms were compared with more robust and accurate but slower algorithms, which yielded similar results. The algorithm proposed by Dr. Højstrup (Hojstrup, 1993) is interesting but is not fundamentally different or superior to those we have tried.

**References**

Akaike, H. (1969). Fitting autoregressive models for prediction. *Annals of the institute of Statistical Mathematics*, 21(1):243–247.

D'Errico, J. (2004). inpaint_nans (http://kr. mathworks. com/matlabcentral/fileexchange/4551-inpaint-nans), matlab central file exchange. *Retrieved*, November:2021.

Hojstrup, J. (1993). A statistical data screening procedure. *Measurement Science and Technology*, 4(2):153.

Højstrup, J. (1999). Vertical extrapolation of offshore wind profiles. In proceedings from EWEA conference in Nice, France.

Kaimal, J. C. and Finnigan, J. J. (1994). *Atmospheric Boundary Layer Flows: Their Structure and Measurement*. Oxford University Press.

Leys, C., Ley, C., Klein, O., Bernard, P., and Licata, L. (2013). Detecting outliers: Do not use standard deviation around the mean, use absolute deviation around the median. *Journal of Experimental Social Psychology*, 49(4):764–766.

Rosner, B. (1983). Percentage points for a generalized ESD many-outlier procedure. *Technometrics*, 25(2):165–172.

---

## Author Comment (AC4)

**Answer to reviewers**

Dear Sir/Madam,

We thank the reviewers for the feedback on our manuscript. Below is our answer to the reviewer's comment.

**Reviewer 2**

**Reviewer's summary:**

The paper "Turbulence in a coastal environment: the case of Vindeby" by Putri et al., provides measurements of turbulence from a field campaign conducted offshore surrounded by complex terrain at Vindeby. The measurements of non-dimensional shear were compared to similarity theory estimates and show reasonably good agreement. Other estimates of co-coherence and turbulence characteristics relevant to offshore wind turbines is also provided. The paper is well structured but there are some aspects of the paper that are not clearly mentioned and could change the result. Reviewing the other comments received for this paper, the two other reviewers have clearly stated some of my concerns as well (regarding flow distortion, computation of the shear, averaging time periods are not clear etc.). In addition to the previous reviewer comments, some additional comments are provided below which would be helpful if clarified in upcoming version. The topic is of interest to the wind energy community and relevant to wind energy science journal.

**Overall response:**

We have re-explored the dataset with respect to the transducer-flow distortion. As highlighted by Reviewer 1 and Dr. Højstrup, flow distortion can be detected in the sonic anemometer records. However, we also found that when we ensemble-average the turbulence characteristics over the sector of interest (220° to 330°), the bias from flow distortion is reduced, provided that a sufficiently high number of samples is considered.

As suggested by Reviewer 1, we have shortened the section discussing the applicability of MOST because of the uncertainties associated with the transducer-flow distortion. We have also reformulated our findings regarding the influence of the wave height on the turbulence characteristics. The detail concerning the flow distortion is addressed in the reply to the community comment by Dr. Højstrup. In addition, we have created appendices to accommodate some of the supporting information that were detailed in the main body of the original submission.

**Comments**
* * *
***Q 2.1*** *Maybe some text related to the initial spectral formulations can be moved to an Appendix. There is nothing new in here, but still relevant to the paper and would reduce the length of the paper.*

**Reply**: We thank the reviewer for the suggestion. We believe that the text related to the spectral formulations is not necessarily be removed to an appendix since it would provide easiness for the reader to understand the results. Furthermore, the length of the text is considerably shortened.
* * *
***Q 2.2*** *Why is friction velocity averaged within the two levels, if you feel the measurements at 6 m are affected by the wave boundary layer? You should revisit this part or provide more justification.*

**Reply**: We used the average at two levels because the friction velocity is usually the parameter associated with the largest uncertainties and the spatial averaging helps reduce them. We revisited this part and we found that the measurements are, to some extent affected by the transducer-induced flow distortion. In the revised manuscript, we carefully review our statements that the measurements at 6 m are affected by the wave boundary layer.
* * *
***Q 2.3*** *Figure 6 shows good agreement in non-dimensional shear when using 6 m measurements with*

*similarity theory. This is confusing and I was under the impression that the friction velocity and z/L were estimated from 18 m and not 45 m. Some consistency/clarification is required here.*

**Reply**: We have clarified in the caption of fig. 6 (fig. 5 in the revised manuscript) that $z/L$ was indeed estimated from the measurements at 18 m.
* * *
***Q 2.4*** *With average significant wave heights (Hs) below 1 m (line 64), and the wave boundary layer*

*is typically 5*Hs, so it would mean most of the time the wave boundary layer is below the lowest measurement height (6 m). There may be instances when the waves can affect the measurements, but this would be very small for such low Hs. Please refer to Hristov et al., 1998 (Wave-Coherent Fields in Air Flow over Ocean Waves: Identification of Cooperative Behavior Buried in Turbulence) for more details on how to assess the impact of the wave boundary layer on measurements. After filtering for nonstationary etc., what is average Hs in Figure 5? Small set of measurements affecting the average shear is somewhat surprising. The deviations observed in MOST are probably not due to the presence of the wave boundary layer impacting measurements, but probably flow distortion. This needs to be investigated further.*

**Reply**: As pointed out by the reviewer, the results in Subsection 5.4 (Subsection 5.2 in the revised submission) indicate that there are only a few instances where we see a clear interaction between the waves and the wind turbulence. The median $H_s$ value was 0.4 m for the period considered, which supports our initial statement that the number of records located in the wave boundary layer is insignificant. We have reformulated the first paragraph of Subsection 5.2, which now reads as " The objective of this subsection is to identify whether the wave-induced turbulence can be detected in the velocity records at 6 m amsl due to the observed turbulence characteristics in Subsection 5.1. Here, the measurements at 6 m amsl are explored in terms of wind-wave interactions, using the wave elevation data collected by the AWR near SMW. A total of 925 high-quality samples collocated in time with the wind velocity data studied were identified. Each wave elevation record was 30 min long and corresponded to a wind direction between 220° and 330°. There exist methods to filter out the wave-induced velocity component from the turbulent velocity component (e.g. Hristov et al. (1998)), but these methods are not addressed herein for brevity."

Following our reply to Reviewer 1 and Dr. Højstrup, we removed fig. 5 from the manuscript as the uncertainties regarding the transducer-induced flow distortion may be too large.

**References**

Hristov, T., Friehe, C., and Miller, S. (1998). Wave-coherent fields in air flow over ocean waves: Identification of cooperative behavior buried in turbulence. *Physical review letters*, 81(23):5245.

---

## Referee Report (RR1)

**Review**

The revised paper "Turbulence in a coastal environment: the case of Vindeby" by Putri et al. analyzes some turbulence characteristics of the marine boundary layer using measurement data from a met mast at Vindeby, which is a site in the Baltic Sea close to shore in confined waters. The authors compare their findings with an empirical model derived from measurements at the met mast Fino 1, which is an offshore site in the North Sea. In particular, the power spectral densities and the co-coherences are investigated for a large range of atmospheric stabilities.

The topic of the consideration of turbulence characteristics in the design of (offshore) wind turbines is of relevance to the wind energy community, especially as turbines grow in size and for the design of floating turbines. The findings are a valuable contribution to this and are therefore in the scope of the wind energy science journal.

Aim to make the paper more focused on its main objective and therefore easier to read and also mitigate some concerns regarding the data analysis and validity of the results.

Reviewing the comments received for the original paper, the replies of the authors to the comments and the revised manuscript, I would like to comment on the readability of the paper, i.e. improving its focus, and on two aspects of the data analysis (flow distortion, computation of the stability measure z/L).

**Improving readability / focusing**

The paper documents the analysis done in great detail. While this is very positive for the aspects of the analysis which provide new insight, it reduces readability when used for aspects which are already known or not in focus of the paper. This has already been mentioned by reviewer 1 to the original manuscript, it has been improved in the revised manuscript, but in my view should be improved further by further focusing only on the most important aspects. These are well summarized in the well written conclusion at the end. Anything which is not needed to come to this conclusion should be removed from the paper or at least be greatly reduced. Some suggestions:

- Chapter 5.1. discusses the applicability of MOST. Figure 5 and the discussion of it is sufficient for this. Figure 6 is not needed – the general distribution of stability classes versus wind speed is known and the number of data points in the stability classes for this particular data set are shown in the graphs later. The discussion of local isotropy and figure 7 are also not needed to conclude the applicability of MOST. They are also not compared to Fino 1 data and not needed for the conclusion or interpretation.
- Chapter 5.2 is very interesting, but the discussion of the influence of the wave boundary layer is far too detailed for the scope of this paper. The conclusion that the effect is not important here can be drawn with a much shorter discussion of a few sentences. I admit that the analysis and the results are very interesting, but for a paper on turbulence in the context of wind energy aiming to compare turbulence parameters with Fino 1 observations this is out of scope. The authors might want to consider to expand this analysis and write a separate paper about it.
- Figure 10, 11, 12 show results for 9 stability classes. However, the conclusion only distinguishes between near neutral, unstable and stable conditions. Also, given the uncertainty in the calculation of z/L and the small number of samples in some of the classes, I would suggest using a maximum of 5 classes. An increase in the number of classes does not lead to new conclusions but increases the scatter and therefore reduces the clarity of the results.

**Comments to the data analysis**

1. *Flow distortion of the sonic anemometers*

Following the comments by Dr Højstrup and the second reviewer, the authors performed a thorough analysis of the presence of flow distortion in their data set and the possible effect of it on their results. The finding are reported in appendix B. They find a clear and relevant (+/- 20%) effect of transducer flow distortion very much in line with the results provided by Dr. Højstrup in his comment. Following this, the authors developed a correction for this effect and recalculated their results including the correction. Comparing figure 10 and B2, the new results show significant differences which proves the relevance of the correction.

However, to my surprise the authors decide not to use the correction. The argumentation in line 209 – 212 seems to be that the error cancels out in the selected wind direction sector. However, this is not shown and is only plausible if the data analysed later are evenly distributed over the wind directions within the sector. It seems very unlikely that this is the case for all of the stability classes analyzed later. Also that "no significant improvement was found for the ensemble-averaged normalized PSD estimates." (line 209) is not a valid argument, since the analysis is done to compare the PSD estimates with the theoretical expectations. The analysis method should therefore not be selected because of its fit to the expectations. Since the error is known, can clearly be found in the data set (figure B1) and can be at least partially corrected for (figure B1), I also do not see any danger of over-processing of the data. Given that the flow distortion clearly has an effect and that this clearly can be reduced by the correction, I recommend to use the corrected data.

Minor comment: Please check the caption of figure B2, there seems to be a copy/paste mistake from the reply to the comment of Dr. Højstrup.

2. *Data processing of the atmospheric stability z/L*

In section 4 the processing of the data for the analysis of the turbulence is well described and a number of quality control measures are taken and explained. However, since for the analysis done later the data are analyzed depending on their stability, a correct assignment of atmospheric stability to each data point is crucial. Selecting the wrong stability class for a data point due to errors or uncertainties in the calculation of the stability parameter z/L can lead to significant changes in the results. However, the processing of the data for the calculation of z/L is not described in chapter 4, there is only a short description in lines 116-122. I suggest to move this paragraph from chapter 3 'Theoretical background' to chapter 4 'Data processing'. Also, the processing steps and especially the quality assurance measures for the fluxes should be discussed. The uncertainty in the calculated z/L data should be investigated, e.g. by comparing the z/L from different heights with each other.

---

## Author Response (AR2)

**Answer to reviewers**

Dear Sir/Madam,

We thank the reviewers for the feedback on our manuscript. Below is our response to the reviewer's comments.

**Reviewer 1**

**Reviewer's summary:**

The revised paper "Turbulence in a coastal environment: the case of Vindeby" by Putri et al. analyzes some turbulence characteristics of the marine boundary layer using measurement data from a met mast at Vindeby, which is a site in the Baltic Sea close to shore in confined waters. The authors compare their findings with an empirical model derived from measurements at the met mast Fino 1, which is an offshore site in the North Sea. In particular, the power spectral densities and the co-coherences are investigated for a large range of atmospheric stabilities.

The topic of the consideration of turbulence characteristics in the design of (offshore) wind turbines is of relevance to the wind energy community, especially as turbines grow in size and for the design of floating turbines. The findings are a valuable contribution to this and are therefore in the scope of the wind energy science journal. Aim to make the paper more focused on its main objective and therefore easier to read and also mitigate some concerns regarding the data analysis and validity of the results.

Reviewing the comments received for the original paper, the replies of the authors to the comments and the revised manuscript, I would like to comment on the readability of the paper, i.e. improving its focus, and on two aspects of the data analysis (flow distortion, computation of the stability measure z/L).

**Improving readability / focusing**

The paper documents the analysis done in great detail. While this is very positive for the aspects of the analysis which provide new insight, it reduces readability when used for aspects which are already known or not in focus of the paper. This has already been mentioned by reviewer 1 to the original manuscript, it has been improved in the revised manuscript, but in my view should be improved further by further focusing only on the most important aspects. These are well summarized in the well written conclusion at the end. Anything which is not needed to come to this conclusion should be removed from the paper or at least be greatly reduced.

**Overall response:**

Generally, the reviewer has recommended to shorten the manuscript to focus only to the points included in the conclusion. We are agree with the reviewer and some of the contents are shortened and omitted in the revised manuscript.

**Suggestions from the reviewer**
* * *
*Q 1.1 Chapter 5.1. discusses the applicability of MOST. Figure 5 and the discussion of it is sufficient*

*for this. Figure 6 is not needed – the general distribution of stability classes versus wind speed is known and the number of data points in the stability classes for this particular data set are shown in the graphs later. The discussion of local isotropy and figure 7 are also not needed to conclude the applicability of MOST. They are also not compared to Fino 1 data and not needed for the conclusion or interpretation.*

**Reply**: We agree with the reviewer to remove Figure 7 and its discussions from the manuscript. On the other hand, Figure 6 is kept in the manuscript, because this figure contains the information of the mean wind speed distribution, which we feel is worthwhile to include for completeness. Furthermore, as suggested by the reviewer, the stability bins are now reduced to 5 only (see Q 1.3), and Figure 6 could demonstrate the relatively poor proportion of the stronger convective cases.
* * *
***Q 1.2*** *Chapter 5.2 is very interesting, but the discussion of the influence of the wave boundary layer*

*is far too detailed for the scope of this paper. The conclusion that the effect is not important here can be drawn with a much shorter discussion of a few sentences. I admit that the analysis and the results are very interesting, but for a paper on turbulence in the context of wind energy aiming to compare turbulence parameters with Fino 1 observations this is out of scope. The authors might want to consider to expand this analysis and write a separate paper about it.*

**Reply**: Section 5.2 plays a key role to demonstrate that the sensor at 6 m is not located in the wave boundary layer during a significant amount of time. We agree to shorten it, but we cannot, unfortunately, remove it completely. This is contrary to to the opinion of the other reviewer who would like us to extend the discussion on the wave effects. We have therefore tried to streamline this topic whilst keeping the salient points in order to satisfy both reviewers.
* * *
***Q 1.3*** *Figure 10, 11, 12 show results for 9 stability classes. However, the conclusion only distinguishes*

*between near neutral, unstable and stable conditions. Also, given the uncertainty in the calculation of z/L and the small number of samples in some of the classes, I would suggest using a maximum of 5 classes. An increase in the number of classes does not lead to new conclusions but increases the scatter and therefore reduces the clarity of the results.*

**Reply**: We agree with the reviewer. To avoid the scatter and to increase the clarity of the results, the stability class is now reduced to 5, for $-0.5 \leq \zeta \leq 0.5$ in Figure 10, 11, 12. However, the stronger convective cases are still mentioned briefly in the manuscript.

**Comments from the reviewer regarding data analysis**
* * *
***Q 1.4 Flow distortion of the sonic anemometers:*** *Following the comments by Dr Højstrup and the*

*second reviewer, the authors performed a thorough analysis of the presence of flow distortion in their data set and the possible effect of it on their results. The finding are reported in appendix B. They find a clear and relevant (+/- 20%) effect of transducer flow distortion very much in line with the results provided by Dr. Højstrup in his comment. Following this, the authors developed a correction for this effect and recalculated their results including the correction. Comparing figure 10 and B2, the new results show significant differences which proves the relevance of the correction. However, to my surprise the authors decide not to use the correction. The argumentation in line 209 – 212 seems to be that the error cancels out in the selected wind direction sector. However, this is not shown and is only plausible if the data analysed later are evenly distributed over the wind directions within the sector. It seems very unlikely that this is the case for all of the stability classes analyzed later. Also that "no significant improvement was found for the ensemble-averaged normalized PSD estimates." (line 209) is not a valid argument, since the analysis is done to compare the PSD estimates with the theoretical expectations. The analysis method should therefore not be selected because of its fit to the expectations. Since the error is known, can clearly be found in the data set (figure B1) and can be at least partially corrected for (figure B1), I also do not see any danger of over-processing of the data. Given that the flow distortion clearly has an effect and that this clearly can be reduced by the correction, I recommend to use the corrected data.*

**Reply**: The differences between Figure 10 and Figure 2.B are significant for stable stratifications or strongly convective conditions only. As suggested by the reviewer, we have removed from the paper the cases where $\zeta > 0.5$ and $\zeta < -0.5$, which are seldom observed in the dataset. For the five remaining stratifications, we confirm that the differences between the corrected and the uncorrected are relatively small.

The reviewer argues that the error on the friction velocity does not cancel out when using sector-averaged values. This statement is contradicted by lines 508-511 in the first version of the revised manuscript. In these lines, we mention that the sector-averaged friction velocity calculated with and without correction differ by 11% or less, which is within the measurement uncertainty. We would like to highlight that the sector selected is relatively narrow (220°-330°), leading to the relatively small difference. In summary, we have decided not to apply the correction for the five selected stability bins since the correction may obscure the results of our analysis.
* * *
**Q 1.5 Minor comment:** *Please check the caption of figure B2, there seems to be a copy/paste mistake from the reply to the comment of Dr. Højstrup.*

**Reply**: The caption of Figure B2 has now been corrected to "Normalised spectra of the along-wind component on SMW with the corrected friction velocity and five stability bins. The red curve is derived from eq. (5) and N denotes the number of samples considered for ensemble averaging".
* * *
**Q 1.6 Data processing of the atmospheric stability z/L:** *In section 4 the processing of the data for the analysis of the turbulence is well described and a number of quality control measures are taken and explained. However, since for the analysis done later the data are analyzed depending on their stability, a correct assignment of atmospheric stability to each data point is crucial. Selecting the wrong stability class for a data point due to errors or uncertainties in the calculation of the stability parameter z/L can lead to significant changes in the results. However, the processing of the data for the calculation of z/L is not described in chapter 4, there is only a short description in lines 116-122. I suggest to move this paragraph from chapter 3 'Theoretical background' to chapter 4 'Data processing'. Also, the processing steps and especially the quality assurance measures for the fluxes should be discussed. The uncertainty in the calculated z/L data should be investigated, e.g. by comparing the z/L from different heights with each other.*

**Reply**: Figure 4 already discusses the momentum fluxes between the different sensors. We feel that a further discussion on $\zeta$ will be repetitive and unnecessary. It also contradicts the objective to shorten the manuscript. Finally studying the height-dependency of $\zeta$ addresses the question of the local similarity theory, which is beyond the scope of the paper.

**References**

**Answer to reviewers**

Dear Sir/Madam,

We thank the reviewers for the feedback on our manuscript. Below is our response to the reviewer's comments.

**Reviewer 2**

**Reviewer's summary:**

I realize that you have performed an extensive revision of the paper. You also made the paper much more concise and short, which is really good. However, this study troubles me because as I think I mentioned in my first review, most of the aspects that you investigate are not really innovative/new. You basically validate the findings from Fino 1 at another site at heights closer to the surface. But since you had measurements much closer to the surface, the innovation (in my opinion) would have been the thorough investigation of the effect of the waves on turbulence. But this later aspect is barely touched. So right now, the study still reads as a technical report and not as a research paper (I am trying to provide with my comments some ideas on how to make it look more like a research paper). My comments are based on the trackchanges version of your revision.

**Overall response:**

We thank the reviewer for the constructive comments. We agree that having measurement close to the sea surface was indeed a great opportunity to study the wave effects on turbulence. However, there are two constraints that the authors face with regard to expanding this part of the analysis. The first is that the first reviewer would like us to remove this part of the manuscript and mainly focus on the analysis of the wind measurements ("the discussion of the influence of the wave boundary layer is far too detailed for the scope of this paper. I admit that the analysis and the results are very interesting, but for a paper on turbulence in the context of wind energy aiming to compare turbulence parameters with Fino 1 observations, this is out of scope"). The second is that the Sea Mast West (SMW) is located in an enclosed sea with limited fetch and as a result, the highest recorded and the median wave heights are 1.5 m and 0.4 m respectively (based on our available data). Wave heights greater that 0.9 m are available only for 60 samples, which mean that it was difficult to draw significant conclusions based on the data availability. This is the reason why we included this limited analysis as a discussion in the paper to highlight the limited wave effects on wind turbulence at this location. In order to satisfy both reviewers, we have tried to streamline this discussion on wave effects but have retained it, in this paper, to show the limited wave effects observed in this location. Again many thanks to the reviewer for their contributions to improving the manuscript and we hope that we have managed to satisfy both, given the constraints outlined above.

**Major comments**
* * *
*Q 2.1 The new abstract reflects my main issue. There is no clear signs of new findings or important*

*conclusions from the analysis of the measurements. Can the message be changed? Could you say that all IEC turbulence and coherence models are not in agreement with offshore measurements? If so, then what is the impact of suchdisagreement? Are important such differences? Well, for that you will need to also apply the Mann model, so perhaps you could do that as well both for the spectra and coherence.*

**Reply**: Firstly, the motivation for this study comes from the lack of rigorous turbulence analysis in the marine atmospheric boundary layer. This is considered to be an ongoing challenge for wind energy science, as summarised in e.g. Veers et al. (2019). Secondly, the present study demonstrates how the datasets from Vindeby and FINO1 complement each other, so the study is not limited to just a comparison. Furthermore, additional novel findings in the study are mentioned which are (1) The scalability of the vertical coherence with a model derived from Bowen et al. (1983) is demonstrated in the marine atmospheric boundary layer. This is a major finding for turbulent loading on wind turbines; (2) The velocity spectra studied on FINO1 and Vindeby shows that for design purposes, using site-specific data, advised in the IEC standards, may be conveniently replaced by a more universal spectral model when the site-specific turbulence data are not available; (3) We demonstrated that, even though the measurements are above the wave sublayer during a significant amount of time, the current definition of the wave sublayer depth $h = 5Hs$ is not conservative; (4) The clear variability of the coherence with the atmospheric stability is highlighted, which calls for a new methodology for the fatigue life design of offshore wind turbine (5) The potential applicability of Klipp's method for diabatic conditions. This is a minor finding for wind energy by more significant for atmospheric science.

The IEC turbulence models are applicable for the ultimate limit state of wind turbines only, where the stratification of the atmosphere is assumed to be neutral. Stating that the IEC turbulence and coherence models are or are not in agreement with offshore measurements would be an oversimplification. The question of the applicability of the IEC turbulence models in the marine atmospheric boundary layer has already been addressed in several studies, including Cheynet et al. (2018) for the coherence and Cheynet et al. (2017) for the one-point spectra. Therefore, the IEC turbulence models are not the main focus of the present study, as they would overlap too much with previous studies.
* * *
***Q 2.2*** *The local isotropy quest. Around line 350 you are evaluating local isotropy. First you mention that the references you provide show that the Sv/Su ratio reaches isotropy easier than Sw/Su. This is not true. With or without correction the Sv/Su ratio is close to 4/3, whereas Sw/Su is never 4/3 unless properly correcting for distortion. So when you say Sw/Su 4/3 ratio is only reached for the 6 m, it is natural to think that the other two sonics need to be corrected. You also say that at 18 m Sw/Su converges towards 1.2 in near-neutral conditions, which is not really shown in the figure (yes the values might reach 1.2 but convergence is perhaps not the right term here). So the message, I think, is not that you need to try to find out whether the u\* estimates are good or wrong (as you now do in an appendix) but to try to get the best time series of your velocity fluctuations as you used these to compute the spectra and coherence as the main results of your analysis (independently whether u\* is good or bad).*

**Reply**: The discussion concerning the local isotropy is now omitted from the manuscript, as suggested by Reviewer 1. The answer to this comment has partly been covered in our initial reply to the reviewer in Q.1.5. In Fig 7, the Sw/Su ratio at 18 m is equal to or above 1.20 for reduced frequency above 5. This can be easily verified using digitalization software or the free online tool WebPlotDigitizer. It should be noted that the ratio 4/3 may not be reached, even without flow distortion, see. e.g. Chamecki and Dias (2004). So the quest for local isotropy is not trivial. The relevancy of studying $u_*$ is addressed in our reply to Dr Højstrup as well as in his short comment.
* * *
***Q 2.3*** *And so then the question is why you kind of reach the value of 4/3 for Sw/Su for the 6 m sonic only, although you have not corrected this? After the coherence analysis you did between the wave velocity and the vertical velocity of the sonic at 6 m you conclude that the waves have a limited impact on the 6-m sonic. So the question is, what would be then the result of this analysis if you have corrected the velocity time series for flow distortion?*

**Reply**: As elaborated in our previous reply to the reviewers, a direct correction of the vertical velocity component needs to rely on wind tunnel tests. The correction discussed in the appendix is on the friction velocity. Following the approach used by Peña et al. (2019), we could argue that if the ratio

$S_w/S_u$ reaches 1.3 in the inertial subrange, no correction would be needed. The short comment by Dr Højstrup suggested that using the ratio alone was not sufficient to guarantee that flow distortion is negligible. Following the suggestions by Reviewer 1, Figure 7 and the associated discussion are now removed from the manuscript (see Q 2.1).
* * *
***Q 2.4*** *The comparison of measurements in Figs. 10-12 with the model by Cheynet et al is really good and it seems one the most important findings. Why is not the model introduced/explained? It seems to be stability dependent so where and how is stability accounted for in the model?*

**Reply**: The description of the model used in Cheynet et al. (2018) would lengthen the manuscript unnecessarily. Therefore, we decided to be more concise and to simply refer to the paper, the post-print of which is publicly available online. The model is indeed stability dependent, where it relies on empirical parameters that were established on FINO1.

**Minor comments**
* * *
***Q 2.1*** *Line 3 delete "(sea . . . MW)"*

**Reply**: We agree with the reviewer. The following "(Sea Mast West/SMW)" is now removed from the manuscript.
* * *
***Q 2.2*** *Line 6 delete "above. . . amsl)"*

**Reply**: "above mean sea level (amsl)" carries an important information for the whole sentence, therefore it is not deleted.
* * *
***Q 2.3*** *Line 11 delete "to some extend"*

**Reply**: Because the flow distortion affects the anemometers up to a certain degree only, therefore the term "to some extent" is not deleted.
* * *
***Q 2.4*** *Line 12 replace "spectrum" to "spectra"*

**Reply**: We agree with the reviewer. The term is "spectrum" is now replaced to "spectra".
* * *
***Q 2.5*** *Line 13 delete "(z/L. . . surface)"*

**Reply**: We agree with the reviewer. The term is now deleted.
* * *
***Q 2.6*** *Line 18 "predictions from FINO1": FINO1 cannot make predictions; predictions are made by models for example.*

**Reply**: We agree with the reviewer. The line is now revised to "The co-coherence of the along-wind component, estimated for vertical separations under near-neutral conditions matches remarkably well with the predictions from the dataset at the FINO1 platform."
* * *
***Q 2.7*** *Lines 20 and 23 dataset or data set. . . use one convention for all entries*

**Reply**: The term "dataset" is selected for the entire manuscript.
* * *
***Q 2.8*** *Lines 25-26 delete them*

**Reply**: The lines are not deleted. We believe that these lines are essential to be kept at the end of the abstract. To improve clarity, the lines are smoothen as "Yet, the dataset recorded at Vindeby and FINO1 covers only the lower part of the rotor of state-of-the-art offshore wind turbines. Further improvements in the characterisation of atmospheric turbulence for wind turbine design will require measurements at heights above 100m amsl".
* * *
**Q 2.9** *Line 39 "non-neutral... than on land": on-land non-neutral conditions are more common, so quite the opposite*

**Reply**: We confirm that non-neutral conditions are likely to be more common above the ocean than on land. For a given mean wind speed at a reference height, the higher roughness on land implies that mechanically-generated turbulence is greater than offshore, leading to a greater Obukhov length. The predominance of diabatic conditions in the MABL has been documented in e.g.Archer et al. (2016). To provide a more accurate information, this line has now been reformulated as "The characteristics of the MABL differ from the overland atmospheric boundary layer (ABL) due to the large proportion of non-neutral atmospheric stability conditions (Barthelmie, 1999; Archer et al., 2016) and low roughness lengths."
* * *
**Q 2.10** *Lines 54-55 "low-frequency fluctuations.... convective conditions". This is not really true. All microscale turbulence models underestimate the low-frequency fluctuations in unstable conditions as they do not account for mesoscale fluctuations. Other models attempt to account for this extending the microscale range they used to cover.*

**Reply**: We agree that there is indeed a strong interaction between mesoscale and microscale atmospheric motion under convective conditions. However, in the present case, the duration of the time series selected is 30 min. Under convective conditions, this duration is likely too short to include mesoscale fluctuations. See for example the study by Smedman-Högström and Högström (1975) which focuses on the time-scale associated with the spectral gap. The reason why microscale models underestimate low-frequency fluctuations in unstable conditions offshore has been discussed in the past. The uniform shear model is not designed for convective conditions, so larger eddies due to buoyancy-enhanced turbulence are not accounted for, as discussed in Chougule et al. (2018). Spectral models designed for unstable conditions are a rarity. They are based on the Minnesota experiment, where the low-frequency fluctuations were high-pass filtered (Kaimal et al., 1976; Drobinski et al., 2004). Finally, many spectral models are based on onshore measurements whereas offshore measurements are known to be associated with greater low-frequency turbulent fluctuations, so onshore models cannot be directly applied to offshore conditions.
* * *
**Q 2.11** *Line 59 delete "in the frequency space"*

**Reply**: Keeping the term "in the frequency space" is crucial here as it directly implies that we do not focus on integral turbulence characteristics but spectral characteristics. Therefore, we decided not to remove this term.
* * *
**Q 2.12** *Line 70 replace "from the established" by ", which are based"*

**Reply**: We agree with the reviewer. The term is now replaced.
* * *
**Q 2.13** *Fig 1-right maybe add a line in the limits of the sector you describe in line 151; and maybe add the distance to Langeland from the mast*

**Reply**: The sector limit is now added in the caption of Figure 1: "Only the wind from 220° to 330° is considered for SMW". The distance of SMW mast to the land is now added in the caption of Figure 2: "SMW is located approximately 1.5 km from Lolland".
* * *
**Q 2.14** *Line 190 around this line, the reader got to know that Davenport's model is applicable to vertical separations, but suddenly Kristiansen and Sacre & Delaunay questioned its applicability to lateral separations? Why they use a model for vertical separations then?*

**Reply**: The sentence mentions Davenport's similarity, not the Davenport model. The Davenport model is the exponential coherence model. The Davenport similarity is the assumption that the coherence (a 2D function) can be reduced to a 1D function by using a non-dimensional frequency $fd/\overline{u}$ where $d$ is a distance.
* * *
**Q 2.15** *Line 199 "This implies that fitting....wind turbines" well this depends on whether you believe more in Bowen than in Davenport and until this point of the manuscript you have not given arguments to believe so*

**Reply**: The argument is elaborated in Cheynet (2019), where the influence of the coherence model on the modal wind loading is explored. We have added a reference to Cheynet (2019) at the end of this sentence.
* * *
**Q 2.16** *Line 208 "Kristiansen & Jensen (1979)... 1/T" so K&J had a similar model to the Bowen model... I mean they seemed to already have given an expression for c3i*

**Reply**: Kristensen and Jensen (1979) discuss only the limits of the Davenport model in terms of turbulence length scales. The coefficient $c_3^i$ was introduced in Cheynet (2019) to modify the Bowen model, based on the discussion from Kristensen and Jensen (1979).
* * *
**Q 2.17** *Lines 246-247 This statement is debatable and you only had a reference to support it*

**Reply**: The statement contains new information, and the one reference we were referring to is a study containing a well-known method. The reference we used to support this statement is the second most cited paper in Boundary-Layer Meteorology since this journal was established in 1970s.
* * *
**Q 2.18** *Lines 331-332 "It should be noted... 18 m amsl": well, yes, the conditions will appear closer to neutral conditions as the friction velocity at 6 m is higher than at 18 m.*

**Reply**: In the present case, the variability of the friction velocity with the height (cf. Fig 4 in the manuscript) is not sufficient to explain the variability of $\zeta$ with the height. The Obukhov length seems to fluctuate relatively little between 6 m and 18 m, whereas the height is three times lower at 6 m than at 18 m. Therefore, $\zeta = z/L$ is much smaller at 6 m than 18 m. The statement that $u_*$ is much larger at 6 m than at 18 m was part of the original manuscript but was discussed in our reply to Dr Højstrup.
* * *
**Q 2.19** *Line 133 and Fig. 5: you do not mention how du/dz is estimated in the computation of $\psi_m$.*

*I think you still need to use a polynomial in the logarithmic of the height to better compute the local gradient at the given height where your z/L value is derived from. What about using the polynomial form of Hoegstroem (1988)? The left and right panels show that du/dz should be computed at the height where z/L is computed. In the text you say that in the right panel you use the friction velocity from 45 m but in the caption you say that z/L is at 18 m.*

**Reply**: We have addressed this comment in a thorough detail in our reply to the reviewer for the original manuscript (Q.1.6).
* * *
***Q 2.20*** *Lines 482-496: I recommend to delete all these lines, they are expendable.*

**Reply**: We believe, on the contrary, that these comments are valuable to illustrate the variability of the spectral gap with the thermal stratification of the atmosphere. These lines directly address the challenge of the flow modelling under a stable stratification, which is relevant for floating offshore wind turbines with low-frequency modes of vibrations.
* * *
***Q 2.21*** *When presenting Fig. 13 you do not talk about the w co-coherence results. There, the Fino 1 model performs quite badly with the three combinations you present*

**Reply**: We agree with the reviewer. Additional lines are added in the manuscript "The vertical co-coherence observed from SMW shows deviates substantially from the one fitted to observations at the FINO platform. The source of such deviations remains unclear."
* * *
***Q 2.22*** *Appendix A: the angle in 629 is between the stresses only (you say stress and wind vector). What will happen if w in particular needs to be corrected?*

**Reply**: The correction used in the present study relies on a multivariate regression for the covariance terms, which are used both on the nominator and denominator. So, in that case, the angle value is unlikely to change significantly. Also, these angles are calculated based on sector-averaged covariance terms, which were observed to differ by less than 11% from the corrected values. A simple sensitivity study can show that using angular value without decimal (as we did) is adequate for the present study.
* * *
***Q 2.23*** *Appendix B: you test your "correction" by looking at the u-spectrum, but the corrections affect the w fluctuations mostly, so I do not understand your idea.*

**Reply**: As elaborated in Appendix B, the correction is for the friction velocity because a direct correction of the *w* component would require specific wind tunnel tests. Since the friction velocity is used as scaling velocity, it affects every velocity component. In appendix B it is also mentioned that in the present study, using sector-averaging (220°-330°) helps reduce the influence of flow distortion on the spectral flow characteristics.

**References**

Archer, C. L., Colle, B. A., Veron, D. L., Veron, F., and Sienkiewicz, M. J. (2016). On the predominance of unstable atmospheric conditions in the marine boundary layer offshore of the us northeastern coast. *Journal of Geophysical Research: Atmospheres*, 121(15):8869–8885.

Barthelmie, R. J. (1999). The effects of atmospheric stability on coastal wind climates. *Meteorological Applications: A journal of forecasting, practical applications, training techniques and modelling*, 6(1):39–47.

Bowen, A. J., Flay, R. G. J., and Panofsky, H. A. (1983). Vertical coherence and phase delay between wind components in strong winds below 20 m. *Boundary-layer meteorology*, 26(4):313–324.

Chamecki, M. and Dias, N. (2004). The local isotropy hypothesis and the turbulent kinetic energy dissipation rate in the atmospheric surface layer. *Quarterly Journal of the Royal Meteorological Society: A journal of the atmospheric sciences, applied meteorology and physical oceanography*, 130(603):2733–2752.

Cheynet, E. (2019). Influence of the measurement height on the vertical coherence of natural wind. In *Conference of the Italian Association for Wind Engineering*, pages 207–221.

Cheynet, E., Jakobsen, J., and Reuder, J. (2018). Velocity spectra and coherence estimates in the marine atmospheric boundary layer. *Boundary-Layer Meteorology*, 169(3):429–460.

Cheynet, E., Jakobsen, J. B., and Obhrai, C. (2017). Spectral characteristics of surface-layer turbulence in the north sea. *Energy Procedia*, 137:414–427.

Chougule, A., Mann, J., Kelly, M., and Larsen, G. (2018). *Simplification and Validation of a Spectral-Tensor Model for Turbulence Including Atmospheric Stability*. Boundary-Layer Meteorology, 167(3):371–397.

Drobinski, P., Carlotti, P., Newsom, R. K., Banta, R. M., Foster, R. C., and Redelsperger, J.-L. (2004). The structure of the near-neutral atmospheric surface layer. *Journal of the atmospheric sciences*, 61(6):699–714.

Kaimal, J., Wyngaard, J., Haugen, D., Coté, O., Izumi, Y., Caughey, S., and Readings, C. (1976). Turbulence structure in the convective boundary layer. *Journal of Atmospheric Sciences*, 33(11):2152–2169.

Kristensen, L. and Jensen, N. (1979). Lateral coherence in isotropic turbulence and in the natural wind. *Boundary-Layer Meteorology*, 17(3):353–373.

Peña, A., Dellwik, E., and Mann, J. (2019). A method to assess the accuracy of sonic anemometer measurements. *Atmospheric Measurement Techniques*, 12(1):237–252.

Smedman-Högström, A.-S. and Högström, U. (1975). Spectral gap in surface-layer measurements. *Journal of Atmospheric Sciences*, 32(2):340–350.

Veers, P., Dykes, K., Lantz, E., Barth, S., Bottasso, C. L., Carlson, O., Clifton, A., Green, J., Green, P., Holttinen, H., et al. (2019). Grand challenges in the science of wind energy. *Science*, 366(6464).

---

## Author Response (AR3)

**Answer to reviewers**

Dear Sir/Madam,

We thank the reviewers for the feedback on our manuscript. Below is our response to the reviewer's comments.

**Reviewer 1**

**Reviewer's summary:**

Dear authors,

This new revised version is much shorter and much more concrete than the original one and this is appreciated. I think that, as you mention, your analysis nicely complements that of Fino1. However, as I mentioned in repeated times, the great value of the analysis in my opinion would have been the exploration of the sea interaction on the turbulence characteristics at the lowest sonic or the sonics. You touched the subject but you quickly disregard its importance at Vindeby. I also think that the analysis is not relevant for offshore wind turbines (as you still try to portrait it) as the levels are nowadays far from those of modern turbines; however I think the analysis is very relevant for marine boundary layer and that it surely may benefit meteorological studies over the sea and wind engineering. I would have stressed this instead of connecting it to the turbines per se.

But since you insist on connecting these findings to wind turbines, then I do not understand why you choose to compare the u co-coherence only with the Davenport-like coherence model only (from the IEC standard models). You can only do it for u using Davenport's model, so why not doing the comparison with the Mann coherence from which you can get also the coherence of v and w? The Mann model has the advantage of describe these 3D spatial variations of velocity fluctuations and it is nowadays very much used for load validation on turbines, bridges and structures. I think such a comparison would give the reader comprehension of the need (or not) to derive better turbulence models.

**Overall response:**

We thank the reviewer for the constructive comments. The main purpose of the present study is to discuss the turbulence spectra from the measurements at Vindeby Wind Farm (at heights ranging from 6 to 45 m above sea level) and to compare them with the measurement at FINO1 (Cheynet et al., 2018) (heights ranging from 40 to 80 m above sea level). This is in line partly with the reviewer's statement "however I think the analysis is very relevant for marine boundary layer and that it surely may benefit meteorological studies over the sea and wind engineering".

We acknowledge that the data at 6 m means a good opportunity to investigate further the wind-wave interactions. We discussed the topic briefly and decided not to disregard the topic. Since the main purpose of this manuscript is to investigate the characteristics of the marine atmospheric boundary layer (MABL) turbulence, we decided not to touch deeper on the topic. Moreover, from the available Vindeby dataset, we did not observe the wind-wave interaction of the magnitude reported and as what was described by Kondo et al. (1972), for instance.

We are fully aware that the Vindeby dataset is located below the recent wind turbines' sizes, but this does not mean that the dataset may not be useful for wind turbine design purposes. In fact, the widely-known Kansas spectra are based on the observations at heights not exceeding 30 m (Kaimal et al., 1972) and have been adopted in the IEC 61400 (IEC 61400-1, 2005). From the Vindeby dataset, we found some similarities with the Kansas spectra for near-neutral conditions. For non-neutral conditions, the turbulence spectra at height 45 m above sea level have a consistent behaviour with the predicted spectra at FINO1 at 41.5 m which do not vary much with the observations at 61.5 m and 81.5 m above

sea level (Cheynet et al., 2018). We believe these conclusions from the Vindeby dataset are worthy scientific findings relevant for both the marine boundary layer characterisation and the design of large wind turbines.

Regarding the coherences, the co-coherence of the along-wind component for a near-neutral condition from the Vindeby dataset is predicted most accurately with the modified Bowen model (Cheynet, 2019). It was shown by Cheynet (2019) that the uniform shear model (Mann model) (Mann, 1994) did account for neither the influence of measurement height nor the presence of the surface. These lead to an overestimation of the co-coherence of both the along-wind and the vertical wind components for near-neutral conditions. This is mentioned in the present manuscript in lines 43-44 "Secondly, the vertical coherence of turbulence is not always described accurately by the spectral tensor (Mann, 1994; Cheynet, 2019)".

**Minor comments**
* * *
***Q 1.1*** *line 5 remove "(amsl)"*

**Reply**: The term "(amsl)" is now removed from the manuscript.
* * *
***Q 1.2*** *line 6 remove "to some extent"*

**Reply**: The term "to some extent" is now removed from the manuscript.
* * *
***Q 1.3*** *line 12 "predictions from the dataset": a dataset cannot make predictions*

**Reply**: The term "predictions from the dataset" is now changed to "results from the dataset".
* * *
***Q 1.4*** *You need to introduce c1i and c2i afer Eq. 13*

**Reply**: The two variables are now introduced after Eq. 13 as "where $c_1^i$ and $c_2^i$ are constants".
* * *
***Q 1.5*** *Line 197 c3i has units so it cannot be just zero*

**Reply**: The statement "For such structures, assuming $c_3^i \approx 0$ may no longer be appropriate." does not imply that $c_3^i = 0$. This line emphasises that the co-coherence does not necessarily converge toward unity as the frequency becomes close to zero (for a small separation distance compared to a typical turbulence length scale).
* * *
***Q 1.6*** *Similar to point 5 in line 428 some units are missing*

**Reply**: Indeed that the $c_3^i$ coefficients in line "The decay coefficients used were, therefore, $[c_1^u, c_2^u, c_3^u] = [6.0, 17.8, 0.02]$ and $[c_1^w, c_2^w, c_3^w] = [2.7, 4.0, 0.16]$ as well as $[c_1^v, c_2^v, c_3^v] = [0, 23.0, 0.09]$" have a unit of an inverse of time. However, the physical interpretation of $c_3^i$ is that the eddy size is limited in the vertical direction, as quoted from Kristensen and Jensen (1979), "If we assume that $D$ is much smaller than a scale of the turbulence $L$, the exact behavior of the spectrum at wavenumbers $K << 1/L$ is not important since the coherence on these wavenumbers is unity." (where $D$ is a separation distance). Therefore, the "unit" for $c_3^i$ is not necessarily shown.
* * *
***Q 1.7*** *I think some of the plots, like that in Figure A1 have units in italics*

**Reply**: The italic units in Fig. 11 and Fig. A1 have now been corrected.

**References**

Cheynet, E. (2019). Influence of the measurement height on the vertical coherence of natural wind. In *Conference of the Italian Association for Wind Engineering*, pages 207–221.

Cheynet, E., Jakobsen, J., and Reuder, J. (2018). Velocity spectra and coherence estimates in the marine atmospheric boundary layer. *Boundary-Layer Meteorology*, 169(3):429–460.

IEC 61400-1 (2005). Iec 61400-3 wind turbines part 1: Design requirements.

Kaimal, J. C., Wyngaard, J. C. J., Izumi, Y., and Coté, O. R. (1972). Spectral characteristics of surface-layer turbulence. *Quarterly Journal of the Royal Meteorological Society*, 98(417):563–589.

Kondo, J., Fujinawa, Y., and Naito, G. (1972). Wave-induced wind fluctuation over the sea. *Journal of Fluid Mechanics*, 51(4):751–771.

Kristensen, L. and Jensen, N. (1979). Lateral coherence in isotropic turbulence and in the natural wind. *Boundary-Layer Meteorology*, 17(3):353–373.

Mann, J. (1994). The spatial structure of neutral atmospheric surface-layer turbulence. *Journal of fluid mechanics*, 273:141–168.

---

## Author Response (AR4)

**Answer to the Associate Editor**

Dear Sir/Madam,

We thank the associate editor for the decision and the constructive recommendations on our manuscript. Below is our response to the associate editor's comments.

**Associate Editor's summary:**

Dear authors. thank you for your revised manuscript.

I suggest that you integrate into the manuscript some parts of the answers that you gave to the refeee:

- "It was shown by Cheynet (2019) that the uniform shear model (Mann model) (Mann, 1994) did account for neither the influence of measurement height nor the presence of the surface. These lead to an overestimation of the co-coherence of both the along-wind and the vertical wind components for near-neutral conditions."

- "The Vindeby dataset is located below the recent wind turbines' sizes, but this does not mean that the dataset may not be useful for wind turbine design purposes. In fact, the widely-known Kansas spectra are based on the observations at heights not exceeding 30 m (Kaimal et al., 1972) and have been adopted in the IEC 61400 (IEC 61400-1, 2005). From the Vindeby dataset, we found some similarities with the Kansas spectra for near-neutral conditions. For non-neutral conditions, the turbulence spectra at height 45 m above sea level have a consistent behaviour with the predicted spectra at FINO1 at 41.5 m which do not vary much with the observations at 61.5 m and 81.5 m above sea level (Cheynet et al., 2018)."

**Overall response:**

We have now incorporated the following into the manuscript:

- "Secondly, the version of the uniform-shear model (Mann, 1994) used within the field of wind energy does not account for the blocking by the ground, which may lead to an overestimation of the co-coherence of both the along-wind and the vertical wind components for near-neutral conditions (Cheynet, 2019). Thus, the co-coherence modelling using the uniform-shear model (Mann, 1994) is not discussed further in the present study." that replaces the lines 43-44 "Secondly, the vertical coherence of turbulence is not always described accurately by the spectral tensor (Mann, 1994; Cheynet, 2019).".

- "Although the Vindeby dataset is located below the recent wind turbines' sizes, this does not mean that the dataset may not be useful for wind turbine design purposes. In fact, the widely-known Kansas spectra were based on the observations at heights not exceeding 30 m (Kaimal et al., 1972) and have been adopted in the IEC 61400-1 (IEC 61400-1, 2005). From Vindeby dataset, we found some similarities with the Kansas spectra for near-neutral conditions. For non-neutral conditions, the turbulence spectra at height 45 m amsl have a consistent behaviour with the predicted spectra at FINO1 at 41.5 m which do not vary much with the observations at 61.5 m and 81.5 m amsl (Cheynet et al., 2018). The comparison between the turbulence characteristics at Vindeby and FINO1 is therefore valuable to further develop comprehensive spectral turbulence models that are suitable for modern OWT designs. Nevertheless, future atmospheric measurements at heights up to 250 m amsl are necessary to obtain the knowledge of turbulence characteristics where the surface-layer scaling may no longer be applicable." in lines 437-445.

**References**

Cheynet, E. (2019). Influence of the measurement height on the vertical coherence of natural wind. In *Conference of the Italian Association for Wind Engineering*, pages 207–221.

Cheynet, E., Jakobsen, J., and Reuder, J. (2018). Velocity spectra and coherence estimates in the marine atmospheric boundary layer. *Boundary-Layer Meteorology*, 169(3):429–460.

IEC 61400-1 (2005). Iec 61400-3 wind turbines part 1: Design requirements.

Kaimal, J. C., Wyngaard, J. C. J., Izumi, Y., and Coté, O. R. (1972). Spectral characteristics of surface-layer turbulence. *Quarterly Journal of the Royal Meteorological Society*, 98(417):563–589.

Mann, J. (1994). The spatial structure of neutral atmospheric surface-layer turbulence. *Journal of fluid mechanics*, 273:141–168.

---

## Author Response (AR5)

**Note regarding the submitted manuscript**

Dear Sir/Madam,

We thank the associate editor for the decision on our manuscript. Below are some notes regarding the submitted manuscript for publication.

**File validation remarks:**

For the next revision, I kindly ask you to remove the text part "Copyright statement. TEXT" from page #1 and "Disclaimer. TEXT" from page #25 in *.pdf manuscript version.

**Overall response:**

We have now removed both the text part "Copyright statement. TEXT" from page #1 and "Disclaimer. TEXT" from page #25 in *.pdf manuscript version.

Please also note that the caption for Figure A1 is set for a two-column format i.e. upper and lower figures. If the manuscript shall be published using a single column-format, then the caption for Figure A1 should be "Friction velocity computed using the eddy covariance method with the double rotation method compared with the Klipp method. The left panel considers only $|z/L| \leq 0.1$ and the right panel considers $0.1 < |z/L| \leq 2$."